

**CitcomSVE 3.0: A Three-dimensional Finite Element Software Package for Modeling**
**Load-induced Deformation for an Earth with Viscoelastic and Compressible Mantle**

Tao Yuan[1], Shijie Zhong[1], Geruo A[2]

[1]Department of Physics, University of Colorado, Boulder, Colorado, USA

[2]Department of Earth Sciences, University of California at Irvine, California, USA

E-mail: tao.yuan@colorado.edu



**Abstract.** Earth and other terrestrial and icy planetary bodies deform visco-elastically under various forces.
Numerical modeling plays a critical role in understanding the nature of various dynamic deformation
processes. This article introduces a newly developed, open-source package, CitcomSVE-3.0, which
efficiently solves the visco-elastic deformation of planetary bodies. Based on its predecessor, CitcomSVE-
2.1, CitcomSVE-3.0 is updated to account for elastic compressibility and depth-dependent density, which
are particularly important in modeling horizontal displacement for visco-elastic deformation. We
benchmark CitcomSVE-3.0 against a semi-analytical code for two types of surface loading problems: 1)
single harmonic loads on the surface and 2) the glacial isostatic adjustment (GIA) problem with a realistic
ice sheet loading history (ICE-6G_D), in which an updated version of sea level equations is incorporated.
The benchmark results presented here demonstrate the accuracy and efficiency of this package. CitcomSVE
shows a second-order accuracy in terms of spatial resolution. For a typical GIA modeling with 122-ky
glaciation-deglaciation history, surface horizontal resolution of ~50 km, and time increment of 125 yr, it
takes ~ 3 hours on 384 CPU cores to complete with less than 5% errors in displacement rates.



## 1. Introduction

Observations and interpretations of solid Earth's displacement and deformation in response to surface loadings and tidal forcing are essential in geoscience for at least three important reasons. First, deglaciation on continents and sea level rise as surface loading processes cause uplifts in glaciated continental regions and subsidence of sea floor, respectively. The amount of sea level rise during the deglaciation process critically depends on solid-Earth's response to such surface loading processes (Mitrovica et al., 2001; Peltier, 1998). Second, the dynamics and stability of ice sheets depend significantly on the uplift rate of the underlying bedrock as ice sheets melt (Gomez et al., 2018). This process may play an important role in assessing the fate of West Antarctica ice sheets that have been losing their mass at an alarming rate. Third, modeling solid-Earth's response to surface loading and comparing the model predictions with relevant observations (e.g., deglaciation-induced sea level change and crustal displacements) is the primary way to infer mantle viscosity and rheology (Lambeck et al., 2017; Milne et al., 2001; Peltier et al., 2015) which is essential to studies of mantle dynamics and Earth's evolution (Zhong et al., 2007).

The solid Earth's response to forcing is determined by solving the equations of motion with relevant rheological properties of the mantle and crust. Under the assumption of spherical symmetry in elasticity and viscosity structure (i.e., only 1-D or radial dependence), analytical solutions to the equations of motion are available in spectral or normal mode domains for the displacement, strain and stress (Longman, 1963; Takeuchi, 1950; Wu and Peltier, 1982). However, the Earth's mantle structure has significant lateral variations as demonstrated by seismic imaging studies on both global (Ritsema et al., 2011; French and Romanowicz, 2015; Tromp, 2020) and regional (e.g., Lloyd et al., 2020) scales. Because of the large sensitivity of mantle viscosity to temperature, lateral variations in mantle viscosity are expected to exceed several orders of magnitude (e.g., Paulson et al., 2005; Ivins et al., 2023). For the mantle with fully 3-D elastic and viscosity structures, numerical solution methods are required to solve the equations of motion. The necessity for numerical solution methods has become increasingly more evident as more observations

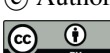



of higher quality (e.g., Bevis et al., 2012) become available to place constraints on the models. In recent
years, numerous numerical methods have been developed, including a spectral-finite element (Martinec,
2000; Klemann et al., 2008; Bagge et al., 2021), finite element (Zhong et al., 2003, 2022; A et al., 2013;
Paulson et al., 2005), finite volume (Latychev et al., 2005), and coupled spectral-finite element (Wu, 2004;
Van Der Wal et al., 2013; Huang et al., 2023) methods.

The CitcomSVE package is a finite element modeling package for solving load-induced

viscoelastic deformation problems in a 3-D spherical shell, a spherical wedge or a Cartesian domain.
CitcomSVE solves the sea level equation and incorporates the effects of polar wander and apparent motion
of the center of the mass (Zhong et al., 2003, 2022; A et al., 2013; Paulson et al., 2005). CitcomSVE works
for 3-D viscoelastic mantle structures with either linear or non-linear viscosity. It works efficiently on
massively parallel computers (>6,000 CPU cores), making it feasible for routine high-resolution GIA
modeling calculations (~30 km horizontal resolution on the Earth's surface and ~10 km vertical resolution
in the upper mantle). CitcomSVE, developed over the last two decades, has been used in GIA studies for
both the incompressible (Zhong et al., 2003, 2022) and compressible (A et al., 2013) mantle with
temperature- (Paulson et al., 2005) and stress-dependent viscosity (Kang et al., 2022), and in tidal
deformation studies for the Moon (Zhong et al., 2012; Qin et al., 2014; Fienga et al., 2024). CitcomSVE
was built from the mantle convection modeling package CitcomS (Zhong et al., 2000, 2008) by replacing
viscous rheology and Eulerian formulation in CitcomS with viscoelastic rheology and Lagrangian
formulation, respectively (Zhong et al., 2003, 2022).

Recently, Zhong et al. (2022) presented an expansive set of benchmark calculations for single

harmonic surface loading, tidal loading, and glaciation and deglaciation loading history (i.e., ICE-6G) for
a significantly improved version of CitcomSVE 2.1. Compared with previous versions of CitcomSVE that
only used 12 CPU cores (e.g., Zhong et al., 2003; A et al., 2013), the most important improvement with
CitcomSVE 2.1 is its capability of efficiently using any large number of CPU cores (e.g., > 6000 CPU cores
as in Zhong et al., (2022)). CitcomSVE 2.1 has also become the first GIA modeling software package that



is open source and publicly available via GitHub (Zhong et al., 2022). However, CitcomSVE 2.1 is for an
incompressible mantle, which limits its applications, especially for studies on GIA-induced horizontal
crustal motions and where realistic elastic structure (e.g., PREM) is necessary (Mitrovica et al., 1994).

This paper presents CitcomSVE 3.0, an extension of CitcomSVE 2.1, by incorporating mantle

compressibility as in A et al. (2013). While the numerical techniques for implementing mantle
compressibility are the same as in A et al. (2013), this paper includes significantly more detailed benchmark
calculations and an improved sea level equation solver. With its public availability via GitHub and efficient
parallel computing, CitcomSVE 3.0 offers the scientific community a powerful computational tool for
solving an important class of geodynamic questions, including the GIA and tidal deformation for Earth's
mantle with realistic viscosity and rheology. The paper is organized as follows. The next section describes
the governing equations for dynamic loading problems and numerical methods. Section 3 defines
benchmark problems and presents benchmark results, including error analyses. Discussions and
conclusions are given in the final section.

## 2. Governing Equations and Numerical Methods

### 2.1. Governing Equations and Viscoelastic Properties of the Mantle

The governing equations for load-induced deformation are derived from the conservation laws of

mass and momentum and Newton's law of gravitation, together with viscoelastic constitutive equation (Wu
and Peltier, 1982; A et al., 2013):

$$\rho_1^E = -(\rho_0 u_i)_{,i}, \tag{1}$$

$$\sigma_{ij,j} + \rho_0 \phi_{,i} - (\rho_0 g u_r)_{,i} - \rho_1^E g_i + \rho_0 V_{a,i} = 0, \tag{2}$$

$$\phi_{,ii} = -4\pi G \rho_1^E, \tag{3}$$

where $\rho_1^E$ is the Eulerian density perturbation, $\rho_0$ is the unperturbed mantle density, $u_i$ represents the
displacement vector with $u_r$ being in the radial direction, $\sigma_{ij}$ is the stress tensor, $\phi$ is the perturbation of



gravitational potential due to deformation, $V_a$ is the applied potential (e.g., rotational and tidal potentials)
when applicable, $g_i$ is the gravitational acceleration with $g = \sqrt{g_i g_i}$, and $G$ is the gravitational constant.
The equations are written in an indicial notation such that $A_{,i}$ represents the derivative of variable $A$ with
respect to coordinate $x_i$, and repeated indices indicate summation.
Both the surface (at radius $r = r_s$) and core-mantle boundary (CMB) ($r = r_b$) experience zero
shear force but are subjected to normal forces
$$\sigma_{ij} n_j = -\sigma_o n_i, \qquad \text{for } r = r_s, \tag{4}$$

$$\sigma_{ij} n_j = (-\rho_c \phi + \rho_c g u_r) n_i, \qquad \text{for } r = r_b, \tag{5}$$

where $\sigma_o$ represents the pressure loads at the surface (e.g., glacial loads) as a function of time and space,
$\rho_c$ is the density of the core, and $n_i$ represents the normal vector of the surface or CMB. The boundary
conditions at the CMB consider the self-gravitational effect for a fluid core (e.g., Zhong et al., 2003). Except
for this CMB boundary condition, the core is not considered explicitly in our numerical formulation. With
such boundary conditions of forces, both the surface and CMB can deform dynamically in both horizontal
and radial directions.
CitcomSVE has implemented formulations for both incompressible (e.g., Zhong et al., 2003; 2022)
and compressible (A et al., 2013) medium. In this study for compressible medium, we follow the
formulation by A et al., (2013). Here, we will only provide a general description for the formulation and
numerical analyses. The details for the compressibility-related topics and numerical analyses of CitcomSVE
can be found in A et al., (2013) and Zhong et al., (2022), respectively. Note that CitcomSVE also
incorporates the effects of polar wander and apparent motion of the center of mass (i.e., degree-1
deformation), and uses a reference frame centered at the center of mass including the mass of loads with no
net rotation of the mantle and crust (Zhong et al., 2022; Paulson et al., 2005; A et al., 2013).





The Earth's mantle is considered as a compressible Maxwell solid, and the constitutive equation
can be written as (e.g., Wu and Peltier, 1982)
$$\dot{\sigma}_{ij} + \frac{\mu}{\eta}\left(\sigma_{ij} - \frac{1}{3}\sigma_{kk}\delta_{ij}\right) = \lambda\dot{\varepsilon}_{kk}\delta_{ij} + 2\mu\dot{\varepsilon}_{ij}, \qquad (6)$$

where η is the viscosity, λ and μ are the Lamé parameters, and $\delta_{ij}$ is the Kronecker delta function. The
strain $\varepsilon_{ij}$ is related to the displacement by $\varepsilon_{ij} = \frac{1}{2}(u_{i,j} + u_{j,i})$. Both Lamé parameters ($\lambda$ and $\mu$) and
viscosity η can be fully 3-dimensional in CitcomSVE models to represent the effects of temperature,
composition and stress on mantle mechanical properties (e.g., Zhong et al., 2003; A et al., 2013; Kang et
al., 2022). However, for this benchmark study, we will only consider radially layered $\lambda, \mu$, and η.

## 2.2. Numerical Analysis

A finite element method is employed in CitcomSVE to solve the governing equations (1)-(3) for
load-induced displacement under boundary conditions (4)-(5) with a Maxwell rheological equation (6)
(Zhong et al., 2003; 2022; A et al., 2013). However, before presenting a weak form of the governing
equations for the finite element analysis, it is necessary to introduce an incremental displacement
formulation, re-formulate the time-dependent rheological equation (i.e., equation 6), and discuss solution
strategies for the gravitational potential that results from mass anomalies associated with mantle
deformation via the Eulerian density perturbation $\rho_1^E$ as controlled by the Poisson's equation (i.e., equation

3).

Define $u_i^n$ and $u_i^{n-1}$ as displacements at times t and t-Δt, respectively, where superscripts *n* and *n-*
*1* represent time steps. Incremental displacement at time *t*, $v_i^n$, is defined as $v_i^n = u_i^n - u_i^{n-1}$ and it is
related to incremental strain $\Delta\varepsilon_{ij}^n$ as
$$\Delta\varepsilon_{ij}^n = \frac{1}{2}(v_{i,j}^n + v_{j,i}^n). \qquad (7)$$



Rheological equation (6) is discretized in time by integrating it from time t-Δt to t, and stress tensor at time
t, $\sigma_{ij}^n$, is given in terms of incremental strain $\Delta\varepsilon_{ij}^n$, stresses at time step *n-1* (i.e., pre-stress), and material
properties as (A et al., 2013; Zhong et al., 2003),

$$\sigma_{ij}^n = \tilde{\lambda}\Delta\varepsilon_{kk}^n\delta_{ij} + 2\tilde{\mu}\Delta\varepsilon_{ij}^n + \tau_{ij}^{pre}, \tag{8}$$

where    $\tau_{ij}^{pre} = (1 - \frac{\Delta t}{2\alpha})/(1 + \frac{\Delta t}{2\alpha})\sigma_{ij}^{n-1} + \frac{\Delta t}{3\alpha}/(1 + \frac{\Delta t}{2\alpha})\sigma_{kk}^{n-1}\delta_{ij}$ ,    $\tilde{\lambda} = [\lambda + (\lambda + \frac{2\mu}{3})\frac{\Delta t}{2\alpha}]/(1 + \frac{\Delta t}{2\alpha})$    ,
$\tilde{\mu} = \mu/(1 + \frac{\Delta t}{2\alpha})$, $\alpha = \eta/\mu$ is the Maxwell time, and $\tau_{ij}^{pre}$ represents the pre-stress at timestep n-1 (A et al.,

2013).

The Poisson's equation for gravitational potential anomaly $\phi$ (i.e., equation 3) is solved in a
spherical harmonic domain for mass anomalies associated with the Eulerian density perturbation $\rho_1^E$ and
the loads (e.g., ice and water loads). For a compressible mantle, $\rho_1^E$ exists throughout the mantle and crust
(see equation 1), and it is necessary to express $\rho_1^E$ at each depth in terms of spherical harmonic degree *l* and
order *m*. The gravitational potential anomaly at radius *r* and time *t* and at degree *l* and order *m*, $\phi_{lm}(r,t)$,
can be related to mass anomalies via Green's function formulation (e.g., A et al., 2013; Zhong et al., 2008).
The solution of $\phi_{lm}(r,t)$ needs to recast to finite element grid points in solving the equation of motion
(i.e., equation 2). It should be pointed out that the transformation for gravitational potential anomalies $\phi$
between the spherical harmonic domain and the spatial domain is computationally rather expensive.
We now present the weak form of the equation of motion (i.e., equation 2) for the compressible
mantle as (A et al., 2013)
$\int_\Omega w_{i,j}[\tilde{\lambda}v_{k,k}\delta_{ij} + \tilde{\mu}(v_{i,j} + v_{j,i})]dV - \int_\Omega \rho_0 g(w_{i,i}v_r + w_r v_{i,i})dV + \sum_l \int_S w_r \Delta\rho_l g v_r dS_l$
$= -\int_\Omega w_{i,j}\tau_{ij}^{pre}dV + \int_\Omega \rho_0 g(w_{i,i}U_r + w_r U_{i,i})dV - \int_\Omega w_{i,i}\rho_0\phi dV$
$+ \sum_l \int_{S_l} w_r(\Delta\rho_l\phi - \Delta\rho_l g U_r + \rho_0 V_a)dS_l - \int_S w_r\sigma_0 dS, \tag{9}$





where integration domain $\Omega$, $S_l$, and $S$ are for the volume, the horizontal surface at some depth with the *l-th*
density boundary, and the Earth's surface, respectively, $w_i$ is the displacement weighting function, $U_i$ is the
cumulative displacements at the previous time step, $V_a$, the applied potential, is only relevant for tidal
loading problems, and $\sigma_0$ is the surface load. Note that the gravitational potential anomalies $\phi$ in equation
(9) depend on unknown incremental displacement $v_i$. We decompose $\phi$ into $\phi = \Phi + \Delta\phi(v_i)$, where $\Phi$ is
the total potential at the previous time step and $\Delta\phi(v_i)$ is the incremental potential determined by $v_i$ and
other incremental mass anomalies at the current time step.
Equation (9) is discretized onto a set of finite element grids to form a system of matrix equations
with unknown vectors of incremental displacement $\{V\}$.
$$[K]\{V\} = \{F_0\} + \{F(\Delta\phi)\},\tag{10}$$

where $[K]$ is the stiffness matrix, $\{F_0\}$ is the force vector representing contributions from the previous time
step, and $\{F(\Delta\phi)\}$ represents contributions from the incremental potential $\Delta\phi$ which depends on the
unknown displacement $\{V\}$ and other incremental mass anomalies. An iteration scheme is applied to
equation (10) to obtain a convergent solution for $\{V\}$ (Zhong et al., 2003).
CitcomSVE was derived from the 3-D finite element code CitcomS for mantle convection in a
spherical shell, and they share many common features including the grid. The spherical shell of the mantle
is divided into 12 caps of similar size, and each cap is further divided into a grid of cells (i.e., elements) of
similar size with eight displacement nodes per element (Zhong et al., 2000; 2008; 2022). This design of
finite element grid is suited for parallel computing, as discussed in Zhong et al., (2008). An important
feature of this grid is its approximately uniform resolution from the polar to equatorial regions (Zhong et
al., 2000; 2003), different from the spectral finite element GIA codes (e.g., Martinec, 2000; Klemann et al.,
2008; Wu, 2004; van der Wal et al., 2013; Huang et al., 2023).



Matrix equation (10) is solved with a parallelized full multigrid method (Zhong et al., 2000; 2008).
The general solution strategy in CitcomSVE follows an iterative scheme that can be summarized as (Zhong
et al., 2003; A et al., 2013):
1) At a given time $t$, $\{F_0\}$ is first evaluated using pre-stress $\tau_{ij}^{pre}$, gravitational potential $\Phi$ and

displacements $U_i$ at the previous time step, $t$-$\Delta t$, and set $\{F\}$ =$\{0\}$.

2) Solve equation (10) using the full multigrid method for incremental displacements $\{V\}$, using $\{F_0\}$

and $\{F\}$.

3) Compute incremental potential $\Delta\phi_{lm}(r,t)$ by solving equation (3) with the incremental

displacements from step 2, and then re-evaluate $\{F\}$. Go back to step 2 to solve for $\{V\}$ again.

4) Repeat steps 2 and 3, until $\{V\}$ converges to a given threshold error tolerance. Then go back to step

1 to march forward in time.

In the implementation of equation (10) in CitcomSVE, all the variables and parameters are
normalized to be dimensionless, and the outputs are also dimensionless. CitcomSVE uses the following
normalization scheme. The coordinates $x_i$ and displacements $u_i$ and $v_i$ are all normalized by the radius of
a planet, $r_s$. The time is normalized by a reference mantle Maxwell time $\alpha = \eta_r/\mu_r$, where $\eta_r$ and $\mu_r$ are
the reference mantle viscosity and shear modulus, respectively. $\eta_r$ is also used to normalize mantle
viscosity and $\mu_r$ is used to normalize elastic moduli, stress tensor and pressure, while the density is
normalized by reference density $\rho_0$. Gravitational potential and centrifugal potential are normalized by
$4\pi G\rho_0 r_s^2$, and the geoid anomalies are normalized by $4\pi G\rho_0 r_s^2/g$. Any other variables can be normalized
by combining the abovementioned scales. However, model input parameters are defined by users as
dimensional values. For example, 3-D mantle viscosity and elasticity models are given by users in separated
files on a regular grid (e.g., 1°x1° grid) at different depths. CitcomSVE reads these parameters from the
files, normalizes them, and interpolates them onto the finite element grids. Along with public releases of
CitcomSVE 2.1 and 3.0 on GitHub, a user manual is available to describe the usage of the code and the
input and output files.



We now finish this section by highlighting the two main differences between incompressible and
compressible models in CitcomSVE (i.e., versions 2.1 versus 3.0). First, the compressible model presented
here does not include the pressure term which is a key component of incompressible models. The absence
of the pressure term simplifies the matrix equation (i.e., equation 10) and its solution procedure, but for the
incompressible model, a two-level Uzawa algorithm is needed to solve for both the pressure and
displacement. Second, mantle compressibility causes mass anomalies or Eulerian density perturbation $\rho_1^E$
throughout the mantle, while for an incompressible mantle, mass anomalies only exist at the surface and
CMB. Consequently, the compressible model is computationally more expensive, particularly for
calculating the gravitational potential anomalies.

## 2.3. Sea Level Change and Sea Level Equation

Understanding and modeling sea level change is important for GIA studies. Sea level change is
controlled by ice volume change and GIA-induced vertical crustal motion and gravitational potential
change. Therefore, the records of sea level change provide essential constraints on GIA processes, including
ice volume change and mantle viscosity. Moreover, sea level change acts as a change of load on the surface,
affecting solid-Earth deformation and gravitational potential. Modeling the GIA processes, one of the major
applications of the CitcomSVE package, requires an accurate sea level equation that describes the sea level
change in this process. A major improvement of CitcomSVE 3.0 over its previous versions is on modeling
sea level changes, and a detailed description is given in this section.
The original sea level equation formulated by Farrell and Clark (1976) provides an elegant way to
incorporate the sea level change into GIA models and can explain the diverging pattern of sea level change
in different regions (e.g., near or far away from former ice sheets). However, the simplified formulation by
Farrell and Clark ignored several factors affecting the accuracy of sea level change modeling. One key
simplification is on the time-dependent ocean-continent function that describes the ocean and continent
distribution, which was assumed to be constant through time in their formulation. The ocean area has varied
by several percent since the last glacial maximum because of the shoreline evolution induced by sea level



rise or fall (Fig. S1). Accounting for the time-dependent ocean-continent function requires modifications
of the sea level equation and affects the predicted sea level change by tens of meters for some regions
compared to that based on Farrell and Clark's formulation (Kendall et al., 2005). Kendall et al. (2005)
provides a modified sea level equation that accounts for the time-dependent ocean function, in which the
variation of ocean area is mainly attributed to two factors: 1) formation or melting of marine ice sheets (i.e.,
ice sheets that lie below sea level), 2) the evolution of shorelines related to the sloping bathymetry and local
sea level change. In previous versions of CitcomSVE, we only considered the variation of ocean function
related to marine ice sheets (A et al., 2013; Zhong et al., 2022). In our new formulation, the sea level
equation is modified to follow the formulation of Kendall et al. (2005). The new sea level equation can be
summarized as follows:
$$L_0(\theta,\phi,t) = [N(\theta,\phi,t) - U(\theta,\phi,t) + c(t)]O(\theta,\phi,t)$$

$$-T_0(\theta,\phi)[O(\theta,\phi,t) - O(\theta,\phi,t_0)] , \qquad (11)$$

Where $t$ is the time with $t_0$ as the initial time (i.e., the onset of loading), $\theta$ and $\phi$ are co-latitude and
longitude, respectively, $L_0$ is the change in sea level relative to the initial stage, $N$ and $U$ are GIA-induced
geoid anomalies and surface radial displacement, $O$ is ocean function (1 for ocean and 0 elsewhere), $T_0$ is
initial topography at $t_0$, and $c$ is introduced for the conservation of water mass and is defined as:
$$c(t) = \frac{1}{A_0(t)}\{-\frac{M_{ice}(t)}{\rho_w} - \int [N(\theta,\phi,t) - U(\theta,\phi,t)]O(\theta,\phi,t)dS$$

$$+\int T_0(\theta,\phi)[O(\theta,\phi,t) - O(\theta,\phi,t_0)]dS\}, \qquad (12)$$

where $M_{ice}$ is the ice mass change relative to the initial stage (i.e., $t_0$), $A_0$ is the ocean area at time $t$, $\rho_w$ is
water density, $N$ and $U$ are relative to $t_0$, and the integral is for the surface of Earth. Following Kendall et
al. (2005), a check for grounded ice is incorporated using the criterion that at any location with
topography $T$ and ice of thickness $I$ and of density $\rho_i$, the ice is considered as ground ice if $I\rho_i > -T\rho_w$.
Only grounded ice is treated as ice load, whereas regions with non-grounded ice (i.e., floating ice) are



treated as oceans. Note that regions with topography $T<0$ and without grounded ice are considered as
ocean where the ocean surface follows the geoid.
The sea level equation can only be solved iteratively because the ocean load associated with sea
level change and ocean function $O(t)$ affect each other, and the unknown initial topography $T_0$ needs to be
determined iteratively to keep the modeled present-day topography consistent with the observed present-
day topography. The algorithm for solving the sea level equation in Kendall et al., (2005) adds an outer
layer of iterations to an otherwise normal GIA modeling that uses pre-determined initial topography $T_0$ and
time-dependent ocean function $O(t)$ to determine $N(t), U(t)$, and $L_0(t)$ for each time $t$ from $t_0$ to the
present day. In the outer layer iteration calculations, at the end of each single complete GIA model run,
time-dependent ocean function $O(t)$ and paleo-topography including initial topography $T_0$ are updated
using newly calculated $U(t)$ and $N(t)$ and the present-day topography, and the updated $T_0$ and $O(t)$ are
then used for next GIA model run. The iteration procedure continues until the initial topography converges.
In practice, the model results would not be altered significantly beyond the second outer iteration. However,
there are noticeable differences in results (e.g., modeled RSL histories) between the first and second outer
iterations for some sites following the algorithm developed by Kendall et al. (2005).
We implemented the algorithm developed by Kendall et al. (2005) in our semi-analytic code (e.g.,
A et al., 2013) and produced consistent results with Kendall et al. (2005). However, running two or three
outer iterations where each iteration is a complete GIA model run of a glacial cycle is computationally
expensive, especially for numerical modeling such as in CitcomSVE, and it would be more efficient if the
results from the first outer iteration (i.e., a single complete GIA model run) can be sufficiently accurate. In
Kendall's algorithm, the time-dependent ocean function $O(t)$ for the first outer iteration is constructed
using fixed shorelines same as that of the present day, except that the extent of oceans may be limited by
the existence of grounded marine ice sheets. However, we found that the first iteration may produce much
improved solutions if $O(t)$ for the first outer iteration is constructed by calculating the change of ocean area
(i.e., ocean-continent transitions) based on ice volume change (i.e., $M_{ice}$) and the present-day topography



(bathymetry), assuming barystatic sea level change on a rigid Earth (i.e., no radial surface displacement).
The ocean function generated in this way generally captures the shoreline evolution for regions experienced
ocean-land transition, and this approximation makes it easy to derive the time-dependent ocean function
for any given ice model. In the next section, we will show the effectiveness of this single outer iteration
method using the improved ocean function in both our semi-analytic solution method and CitcomSVE 3.0.

## 3. Example Calculations and Benchmark Results

Two example problems solved using CitcomSVE 3.0 are presented here. They are: 1) surface
loading problems with a single spherical harmonic in space and step-function (i.e., Heaviside function) in
time; 2) GIA problems with ICE-6G_D ice history model. For each example problem, the elastic and
viscosity structures are chosen to be dependent only on the radius (i.e., 1-D) so that CitcomSVE solutions
can be benchmarked against semi-analytical solutions. The following benchmarks largely follow the
approaches of Zhong et al. (2022).

### 3.1. Surface loading in a single spherical harmonic in space and step-function in time.

#### 3.1.1. Definition of the surface loading problem.

For the first example problem, we consider a surface load $\sigma_0$ (see equation 4) corresponding to
amplitude of topographic variation $d$ with density $\rho_0$ at a single harmonic function in space and step-
function in time:

$$\sigma_0(t,\theta,\varphi) = \rho_0 g d \cos(m\varphi) p_{lm}(\theta) H(t) = \rho_0 g d \bar{P}_{lm}(\theta,\varphi) H(t), \qquad (13)$$

where $H(t)$ is the Heaviside function (i.e., $H(t)=1$ for $t \geq 0$; $H(t)=0$ otherwise) and $\bar{P}_{lm}(\theta,\varphi) =$
$\cos(m\varphi) p_{lm}(\theta)$ is the cosine part of spherical harmonic functions in the real form. Note that only cosine
terms of longitudinal dependence are considered for simplicity. A small amplitude of the load height is used
to avoid large grid deformations. We assume an ocean-free Earth for this example and ignore any sea-level-
related calculations. The density and Lame parameters for lithosphere and mantle are from PREM, except
that for the crust layer those properties are replaced to be same as the underlying mantle, and the viscosity



structure is from VM5a (Peltier et al, 2015). See Table 1 for model parameters. Time-dependent surface 3-
D displacements and gravitational potential anomalies are computed using the newly updated CitcomSVE
and compared with those from semi-analytical solutions (Han and Wahr, 1995; Paulson et al., 2005; A et
al., 2013). The results are presented in terms of load Love numbers $h_l$, $k_l$, and $l_l$ at harmonic degree $l$ for
radial displacement, gravitational potential, and horizontal displacement, respectively. The definitions of
load Love numbers in the context of CitcomSVE calculations are given in equations 37-41 of Zhong et al.,

(2022).

**Table 1. Model parameters for benchmarks**

| Model parameters | value |
|---|---|
| Earth radius $r_s$ | 6371 km |
| CMB radius $r_b$ | 3485.5 km |
| Reference density $\rho_0$ | 4400 kg/m$^3$ |
| Core density | 10895.62 kg/m$^3$ |
| Water density $\rho_w$ | 1000 kg/m$^3$ |
| Ice density $\rho_i$ | 917.4 kg/m$^3$ |
| Reference shear modulus $\mu$ | 1.4305x10$^{11}$ Pa |
| Modified Fluid Love number $k_{2f}(1+\delta)$ | 0.9521091 |
| Mantle reference viscosity $\eta$ | 2x10$^{21}$ Pa·s |
| Gravitational acceleration $g$ | 9.82 m·s$^{-2}$ |
| | |
| VM5A viscosity model: | |
| The surface to 60 km depth | 10$^{26}$ Pas |
| 60 to 100 km depth | 10$^{22}$ Pas |
| 100 to 670 km depth | 4.853x10$^{21}$ Pas |
| 670 to 1170 km | 1.5048x10$^{21}$ Pas |
| 1170 km to CMB | 3.095x10$^{21}$ Pas |





### 3.1.2. Benchmark results.

We have computed a set of model cases using CitcomSVE for four numerical resolutions and six loading harmonics. Four different numerical resolutions of R1-R4 are for 12x(32x32x32), 12x(64x64x64), 12x(64x96x96) and 12x(64x128x128), respectively, where the first number, 12, indicates the number of spherical caps that the spherical surface is divided into, and the subsequent numbers indicate the number of elements in the radial and two horizontal directions in each cap (Zhong et al., 2022). Six different loading harmonics are included for (1,0), (2, 0), (2,1), (4, 0), (8, 4), and (16, 8) where the first and second numbers in parenthesis ($l$, $m$) indicate spherical harmonic degree $l$ and order $m$, respectively. Each case is named by its loading harmonic and numerical resolution; for example, case l2m0_R1 corresponds to the case where the loading harmonic is (2, 0) and the resolution is R1. For l16m8, an additional case with resolution 12x(80x128x128) is included (i.e., l16m8_R5). Each case is computed for 40 Maxwell times (i.e., 40$\alpha$ or non-dimensional time of 40), using a non-dimensional time increment of 0.2. Figure 1 shows $h_l(t)$, $k_l(t)$, and $|l_l(t)|$ for cases with different loading harmonics and numerical resolutions, together with semi-analytical solutions. Table 2 shows both numerical and analytical results of these Love numbers at t=0 and 40 for a selected set of cases (supplementary Table S1 for all the cases). Solutions at t=0 represent the elastic responses of Earth, and the magnitudes of those Love numbers generally increase with time due to viscous relaxation and finally reach nearly stable states after certain time periods (Fig. 1).

**Table 2: Comparison of Load Love Numbers $h_l$, $k_l$, and $l_l$ Between CitcomSVE and Semi-Analytical Solutions**

| Case[a] | $h_l(0)$[b] | $k_l(0)$ | $|l_l(0)|$ | $h_l(40)$ | $k_l(40)$ | $|l_l(40)|$ |
|---|---|---|---|---|---|---|
| l1m0 | -1.2546(-1.2543) | -1.0000(-1.0000) | 0.8864(0.8866) | -1.4968(-1.4964) | -1.0000(-1.0000) | 1.9101(1.9090) |
| l2m0 | -0.9574(-0.9577) | -0.3038(-0.3041) | 0.0203(0.0200) | -2.4066(-2.4066) | -0.9392(-0.9396) | 0.8229(0.8216) |
| l2m1 | -0.3056(-0.3058) | 1.0948(1.0944) | 0.1118(0.1118) | 0.6178(0.6151) | 2.2003(2.1973) | 0.1891(0.1884) |
| l4m0 | -1.0247(-1.0251) | -0.1341(-0.1342) | 0.0569(0.0568) | -4.4395(-4.4402) | -0.9410(-0.9416) | 0.3423(0.3411) |
| l8m4 | -1.2372(-1.2376) | -0.0772(-0.0772) | 0.0303(0.0302) | -8.8084(-8.8405) | -0.9563(-0.9605) | 0.0977(0.0958) |
| l16m8 | -1.6825(-1.6868) | -0.0573(-0.0574) | 0.0228(0.0229) | -17.535(-17.847) | -0.9530(-0.9726) | 0.0435(0.0479) |
| l16m8_R5 | -1.6805(-1.6868) | -0.0572(-0.0574) | 0.0228(0.0229) | -17.623(-17.847) | -0.9579(-0.9726) | 0.0464(0.0479) |



ᵃCase names follow this notation: l1m0 stands for loading harmonic for l=1 and m=0. All CitcomSVE
solutions in this table are for resolution R4 (12x64x128x128), except for l16m8_R5, which has a resolution
R5 (12x80x128x128).
ᵇLoad Love numbers are provided at 0 and 40 Maxwell time. Each entry includes semi-analytical solutions
inside the parentheses and CitcomSVE solutions outside the parentheses.

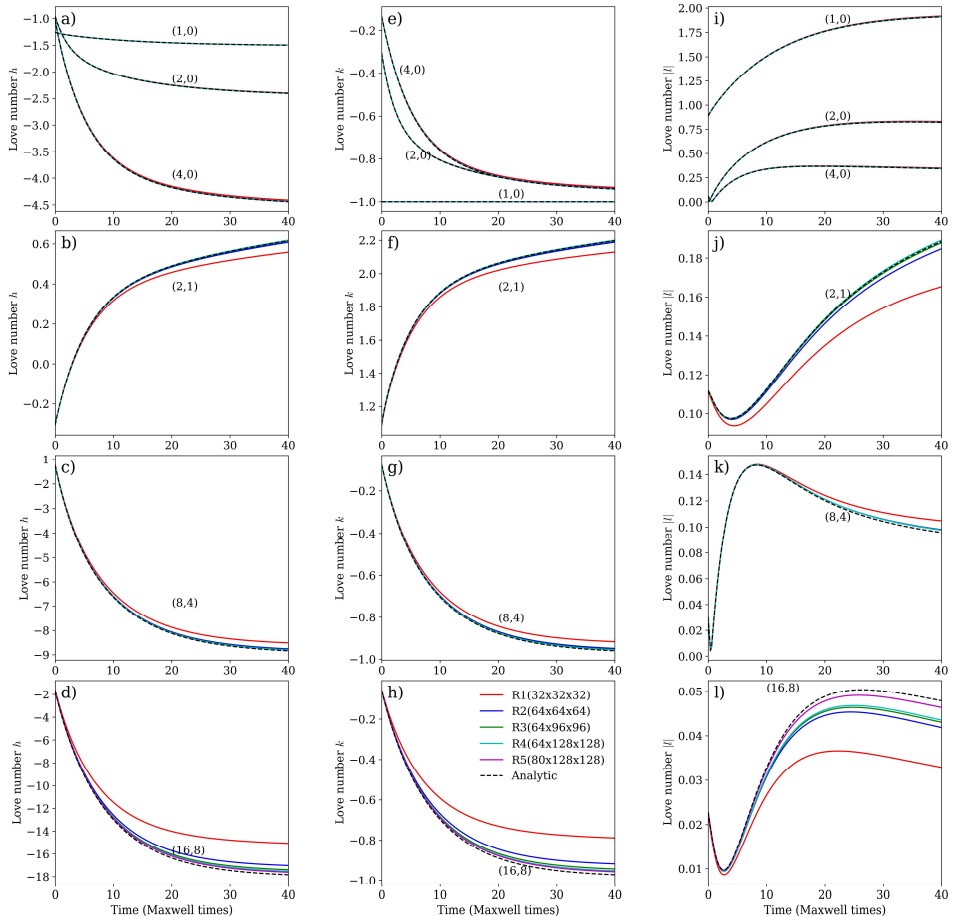


Figure 1. Love numbers $h$, $k$ and $l$ for cases with different loading harmonics from CitcomSVE and
analytical solutions. The first, second, and third columns are for Love number $h$, $k$ and $|l|$ (i.e., the absolute
values of Love number $l$), respectively. The first row is for loading harmonics (1,0), (2,0) and (4,0). The
following rows are for loading harmonics (2,1), (8,4) and (16,8), respectively. Each loading case has
solutions from four different spatial resolutions (R1-R4), except that loading case (16,8) has an additional
calculation with resolution R5.



The comparison shows a good agreement between numerical solutions and semi-analytical
solutions. For long-wavelength loadings (e.g., l1m0, l2m0, and l4m0), numerical solutions at different
resolutions (R1-R4) are nearly identical to semi-analytical solutions, as shown in Figure 1. However, for
l2m1 cases with the polar wander effect, resolution R1 shows significant numerical errors, whereas
calculations with higher resolutions (R2-R4) deliver a remarkable fit to the semi-analytical solution,
suggesting that polar wander is more challenging to compute in numerical models (e.g., Paulson et al., 2005;
A et al., 2013; Zhong et al., 2022). For shorter wavelengths (such as l8m4 and l16m8), low-resolution
numerical results differ noticeably from semi-analytical solutions. As the numerical resolution increases,
the results match the semi-analytical solutions much more closely (Figure 1). For l16m8, case R5
significantly reduces errors in $l_l$ compared to R4. Note that R5 has a higher vertical resolution in the upper
mantle but the same horizontal resolution as R4 (Fig.1 and Table 2). Grid size in the vertical direction is
not uniform since grids get refined vertically in the upper mantle and lithosphere for each model. For cases
with 64 elements in the vertical direction (R2, R3 and R4), the vertical resolutions are about 20 km, 40 km,
and more than 50 km  in the lithosphere, upper mantle and lower mantle, respectively, whereas R5, with a
total of 80 elements in the vertical direction, has a vertical resolution ~ 20 km in the upper mantle. Note
that the load Love number for horizontal displacement is presented as $|l_l(t)|$, because CitcomSVE only
conveniently determines $l_l^2(t)$ (Zhong et al., 2022), although it is possible to determine the $l_l$ based on
vector spherical harmonic decomposition of horizontal surface motion (Wu and Peltier 1982).
We determine numerical errors by computing amplitude and dispersion errors (e.g., Zhong et al.,
2003; A et al., 2013; Zhong et al., 2022). Amplitude error $\varepsilon_a$ and dispersion error $\varepsilon_d$ are computed using
the following equations (Zhong et al., 2022):
$$\varepsilon_a = \frac{\int_0^T |S_n(l_0,m_0,t) - S_{sa}(l_0,m_0,t)| dt}{\int_0^T |S_{sa}(l_0,m_0,t)| dt},$$
(14)

$$\varepsilon_d = \frac{\int_0^T \max [|S_n(l,m,t)|] dt}{\int_0^T |S_{sa}(l_0,m_0,t)| dt},$$
(15)



where $l_0$ and $m_0$ represent the loading harmonic degree and order, $S_n$ and $S_{sa}$ are solutions of load Love
numbers from CitcomSVE and semi-analytical methods, respectively, $T$ is the total model time (i.e., 40),
and in equation (15) for the dispersion error, max represents the maximum value for all the non-loading
harmonic degrees $l$ and orders $m$. The response should only occur at the loading harmonic for the spherically
symmetric mantle structure considered here. Therefore, amplitude error $\varepsilon_a$ measures the accuracy at the
loading harmonic and dispersion error $\varepsilon_d$ measures the accuracy at other harmonics. Note that the errors
defined in equations (14) and (15) are similar to norm-1 errors.

Figure 2 shows the amplitude errors of load Love numbers as a function of horizontal numerical

resolution (i.e., the horizontal grid size ranging from ~200 km to ~50 km at the surface for resolutions R1-
R4) for all cases. For most of the calculations with different loading harmonics, the amplitude errors
decrease with decreasing horizontal grid size with a slope of close to 2 in the log-log plot of Figure 2,
especially for Love numbers $h_l$ and $k_l$. This suggests that the error is roughly proportional to the square of
the grid size, aligning with the expected second-order accuracy for trilinear elements in CitcomS (e.g.,
Zhong et al., 2008). It is worth noting that from R1 to R4, the increase in vertical resolution is not
proportional to the increase in horizontal resolution, which may cause the slope in Figure 2 to deviate from
2. Figure 2 shows that with a horizontal resolution of ~ 50 km, the accuracy of CitcomSVE is better than
0.1% up to spherical harmonics of degree 4 and better than 2% up to spherical harmonics of degree 16 in
terms of Love numbers $h_l$ and $k_l$. For Love number $l_l$, the errors are slightly larger than that for $h_l$ and $k_l$.
Compared to the benchmark results of CitcomSVE 2.1 (Zhong et al., 2022), the errors presented here are
generally larger for cases with the same resolutions, which is understandable considering that CitcomSVE
3.0 solves for models with higher complexity (i.e., the internal density variations caused by compressibility
and density discontinuities).





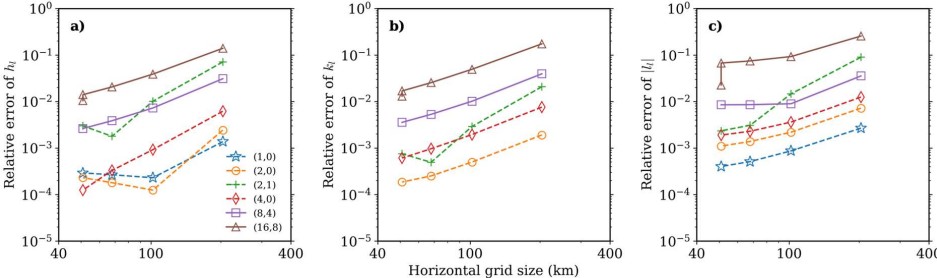


Fig 2. Amplitude errors of Love numbers $h$ (a), $k$ (b) and $l$ (c) as a function of numerical resolutions (i.e., R1-R4, corresponding to horizontal resolutions of approximately 200 to 50 km). For Love number $k$ of loads (1,0), all calculations with different resolutions have a relative error of less than $10^{-5}$ and are not shown in this figure. Note that R4 and R5 have the same horizontal but different vertical resolutions.


## 3.2. Glacial isostatic adjustment using ICE-6G and VM5a

This section presents the benchmark for an example GIA model with ICE-6G and VM5a (Peltier
et al., 2015). A GIA model calculation requires solving governing equations (1)-(3) together with boundary
conditions (4)-(5) and the sea-level equation (11) to determine time-dependent gravitational anomalies and
displacements at the Earth's surface and sea level changes. As discussed in section 2.3, to deal with the
non-linear nature of the sea level equation, multiple iterations of complete GIA model runs may be needed
(Kendall et al., 2005). Before presenting benchmark results for CitcomSVE 3.0 against the semi-analytical
method, we will first demonstrate how the one-iteration solution method discussed in section 2.3 may be
used to achieve adequate accuracy of GIA solutions using the semi-analytical method.
3.2.1. A one-iteration solution method for the sea level equation.
We have implemented the multiple outer iteration algorithm by Kendall et al., (2005) for the sea
level equation in our semi-analytical code (A et al., 2013). For ICE-6G and VM5a, calculation K3 represents
the reference case with convergent solutions after three outer iterations, based on Kendall's original
approach. The normalized ocean area which is a measure of the ocean function $O(t)$ for K3 varies between
~0.66 at the last glacial maximum (LGM) and ~0.71 at 122 kybp and the present-day (Fig. S1). Figure S1
also shows the ocean area after the first outer iteration for calculation K3, which, denoted as K1, differs



significantly from that of K3. Calculation AS1 represents a single outer iteration model run using our pre-calculated ocean function $O(t)$ as discussed in section 2.3, and AS2 represents the results from the second outer iteration after AS1 using the updated ocean functions $O(t)$ and initial topography $T_0$, Figure S1 clearly demonstrates that AS1, different from K1, is very similar to K3 and AS2, while the latter two are identical, indicating that the ocean function $O(t)$ for our first outer iteration (AS1) is a fairly accurate representation of the convergent solutions of the Kendall's original approach (K3). Note that the present-day topography is used as initial topography $T_0$ for calculations AS1 and K1.

Using RSL from K3 as standard results, Fig. S2 shows that the maps of RSL difference (i.e., the accuracy) to K3 from calculations AS1, K1 and AS2 at 5 kybp, 10 kybp and 15 kybp. The absolute error in RSL from AS1 is negligibly small for most regions (Fig. S2a, S2d and S2g), whereas the absolute error from K1 is much worse, especially at 20 kybp (Fig. S2h). AS2 is identical to K3, the standard results (Fig. S2c, S2f and S2i). Admittedly, there are relatively large errors in some localized regions for AS1, such as Hudson Bay and the Arctic Ocean near Fennoscandia for some periods (Fig. S2a and S2d), because we ignore the change in surface radial displacement when deriving the pre-calculated ocean function used in AS1. However, the largest errors in those areas mostly occur in the ocean, while along the coastlines where paleo-relative sea level records are available, the absolute errors are all less than 10 meters (Fig. S2a and S2d). Figure S3 shows the modeled RSL curves at four representative sites including Hudson Bay and Fennoscandia from K3, K1, AS1 and AS2 calculations. The results are consistent with that from Figure S2 in that the errors in modeled RSL from AS1 (i.e., the single outer iteration model run using our revised method for ocean functions) are negligible, whereas the errors from K1 are evident, especially for far-field sites. Note that even at Churchill, which is on the coastline of Hudson Bay, AS1 has negligible errors in RSL calculations.

To further assess the errors in RSL from our AS1 model, we tested two additional GIA calculations with extremely strong or weak mantle viscosity models. For both cases, the lithospheric thickness is 100 km. For the strong mantle case, the entire mantle below the lithosphere has a viscosity of $5\times10^{22}$ Pas. For



the weak mantle case, the 200 km thick asthenosphere below the lithosphere and the rest of the mantle have
viscosities of $5 \times 10^{18}$ Pas and $10^{20}$ Pas, respectively. Figure S4 shows similarly small errors for both cases
to that of VM5a (Fig. S2), indicating the reliability of our AS1 model.

Other pre-calculated ocean functions $O(t)$ for any given ice model may be constructed to obtain

more accurate RSL results in our AS1 method by replacing the "rigid Earth" approximation with others,
for example, the isostasy approximation in which surface elevation changes to compensate the surface
loads. Another possible way is to perform a full GIA modeling with three outer iterations (i.e., for outer
iterations to converge) for a reference viscosity model and use the ocean functions from the last outer
iteration as the pre-calculated ocean functions for any other GIA calculations with reasonable viscosity
models in our AS1 method. We test such a strategy by using a reference viscosity model which has a 100-
km thick elastic lithosphere and its underlying mantle with a uniform viscosity of $10^{21}$ Pas and then applying
the resulting pre-calculated ocean functions for those same two GIA cases with extremely strong or weak
viscosity models as in Figure S4. The resulting errors in RSL for those two cases (Fig. S5) are similar to
that in Figure S4 for which the "rigid Earth" approximation was used in building the pre-calculated ocean
functions.

To quantify the upper bound of errors in RSL by using one outer iteration (e.g., our AS1 method),

we compute 806 GIA models covering a wide range of mantle viscosities and determine RSL histories for
a large number of sites in three regions including North America, Fennoscandia, and far fields using both
AS1 and K3 methods. The numbers of sites are 18, 12, and 36 for North America, Fennoscandia, and far
fields, respectively. The North American and Fennoscandian sites are from Peltier et al., (2015), and the
far-field sites are from Lambeck et al., (2014). These models, same as those in Kang et al., (2024), have
three viscosity layers: a lithosphere of 100 km thick, the upper and lower mantles, and use ICE-6G_D as
the ice history (Peltier et al., 2015, 2018). The viscosity varies from $10^{19}$ Pas to $10^{21.5}$ Pas in the upper
mantle and from $10^{20.5}$ Pas to $10^{23.5}$ Pas in the lower mantle. The relative error (i.e., the relative difference



from the reference case K3) in modeled RSL for each site is defined as $\epsilon_i = \frac{\int_0^T |RSL_{x,i}(t) - RSL_{K3,i}(t)| dt}{\int_0^T |RSL_{K3,i}(t)| dt}$, where
$RSL_{x,i}$ is the modeled RSL at site $i$ for case K1, AS1, or AS2, $RSL_{K3,i}$ is for the reference case K3, and the
integral is for the total model time duration. The regionally averaged relative error $\epsilon$ is defined as the
average error among all sites within each region, i.e., $\epsilon = \Sigma\epsilon_i / N$, where $N$ is the total number of sites within
each region. The maximum regionally averaged relative error among those 806 GIA models is less than 5%
(Supplement Table 2) for our AS1 method.

We also quantify the maximum absolute error in RSL, defined as the maximum of $|RSL_x(t) -$

$RSL_{K3}(t)|$ among all time periods $t$ and all sites in each region from those 806 calculations (Supplement
Table 2). For far-field sites where RSL is mainly controlled by ocean functions and ice volume changes,
the maximum absolute error in RSL is less than 3 meters for the AS1 method but more than 10 meters for
the K1 method, consistent with Fig. S1 in that AS1 provides more accurate ocean functions than K1.
However, the maximum absolute error in near-field RSL is more significant and up to ~23 meters for both
AS1 and K1 methods, reflecting the fact that near-field ocean functions and paleo-topography are more
affected by visco-elastic deformation. Fig. S6 shows the RSL curves for the site and viscosity model
corresponding to the maximum absolute error of ~23 meters in RSL for AS1. Note that at the site for this
case with the maximum absolute error, the total RSL change exceeds 600 meters and the RSL from AS1 is
not significantly different from that from K3 (Fig. S6). Depending on factors including the user's goal, RSL
data quality, and requirements for accuracy and efficiency of GIA calculations, AS1 could be a viable
method to obtain reliable RSL in both far fields and near fields with minimal computational cost.

We summarize our attempts to get accurate RSL results from a single complete GIA model run as

follows. Since the purpose of multiple outer iterations is to update ocean function history and initial
topography successively to be consistent with the present-day topography and a given ice model (Kendall
et al., 2005), our strategy is to construct pre-calculated ocean functions and initial topography that would
lead to RSL solutions with an adequate level of accuracy with a single complete GIA model run (i.e., the



AS1 method). The present-day topography would be a good approximation for initial topography if a model
starts with an ice-sheet distribution similar to that of the present day (i.e., the interglacial period), as in the
benchmark study here. We found that three outer iterations of complete model runs with successively
updated ocean functions and initial topography could be replaced with our AS1 method, depending on
users' goals and requirements for the error levels. For example, studies on global properties of RSL could
achieve adequately accurate results from one single complete run (i.e., AS1) with properly constructed pre-
calculated ocean functions, as we discussed. If the goal is to model the RSL for one particular near-field
site as accurately as possible, it would be more prudent to run two or three outer iterations of complete GIA
runs with successively updated ocean functions and initial topography following Kendall et al. (2005). It is
worthwhile to mention that, when modeling RSL changes, one should also consider other factors including
the errors in RSL records (often exceeding 10 m in near field during the rapid deglaciation (Peltier et al.,
2015; Lambeck et al., 2017)), the relatively low resolution of global ice models, inherent numerical errors,
and unaccounted processes in the current sea level equation (e.g., erosion and sedimentation).

Our above-mentioned results are particularly relevant for numerical modeling given its

computational cost. CitcomSVE 3.0 fully supports the multiple outer iteration approach using pre- and post-
processes to update ocean functions and initial topography. In the following GIA benchmark, we compare
the results from a single complete CitcomSVE  model run with our semi-analytic solutions of the first outer
iteration (i.e., AS1), using the pre-calculated ocean functions constructed by assuming the "rigid Earth" and
the present-day topography as the initial topography. This comparison ensures that CitcomSVE and semi-
analytic calculations have the same ocean functions and initial topography, such that the differences in
solutions between CitcomSVE and semi-analytical methods are solely related to numerical errors rather
than differences in the models.
3.2.2. Definition of the GIA problem.

Since one of the most important applications for CitcomSVE is to model the GIA processes, it is

essential to perform a benchmark with glaciation-deglaciation history as surface loads, considering the



effects of polar wander, apparent center of mass motion and ocean loads determined by the sea-level
equation. Note that the same type of benchmark has been published for the incompressible version
CitcomSVE 2.1 (Zhong et al., 2022), and we largely follow the setups of that previous work except that the
current calculations consider mantle compressibility (i.e., the PREM model), and that the updated sea level
equation is used as discussed in the last sub-section (i.e., the AS1 method). The Earth model used in this
case is the same as the one used for single harmonic loading examples in the previous section.

In this case, the surface load consists of a full glaciation-deglaciation cycle, based on the ICE-6G_D

ice model (Peltier et al., 2015, 2018) that includes the last 122 thousand years from the last interglacial
period to the present day. We assume that Earth was in an equilibrium state at the onset of loading (i.e., 122
kybp), and that the surface displacements and gravitational potential anomalies since 122 kybp are induced
by ice height variations relative to the initial stage and the corresponding change in ocean loads. We
computed six cases using CitcomSVE 3.0 with different spatial-temporal resolutions and cut-off values for
the maximum spherical harmonic degrees used in calculating gravitational potential (Table 3). Cases
GIA_R1, GIA_R2, and GIA_R3 have spatial resolutions of 135 km, 81 km, and 50 km (i.e., a total number
of elements of  12x48x48x48, 12x48x80x80, and 12x64x128x128), respectively, and a temporal resolution
of 125 years per step. Case GIA_R3_LT is the same as GIA_R3 except with a longer time increment of
250 years per step before LGM (i.e., 26 kybp). Cases GIA_R3_LT_SH20 and GIA_R3_LT_SH64 have a
cut-off value of 20 and 64 for the maximum spherical harmonic degrees, respectively, compared to 32 for
other cases. Note that same as CitcomSVE 2.1 (Zhong et al., 2022), computing gravitational potential in
the spherical harmonic domain can be computationally expensive. On the other hand, the semi-analytical
solution is obtained using spherical harmonic degrees and orders up to 256.

It should be noted that in the current implementation, CitcomSVE reads in ice loads defined on

regular grids (e.g., 1ºx1º grid) and then interpolates the loads to the irregular finite element grids, whereas
semi-analytical calculations use spherical harmonic expansions of ice loads to a maximum spherical
harmonic degree and order (i.e., 256 in this study) as inputs. The interpolation may cause inconsistent



representations of ice loads between CitcomSVE and the semi-analytical calculations. To understand the
potential error resulting from the interpolation, we test another case GIA_R3B, which is the same as
GIA_R3 except that, for this case, we let CitcomSVE read in ice loads that are computed on CitcomSVE
finite element grid points from summing up all the spherical harmonics as used for the analytical solutions,
thus avoiding the interpolation from the regular grids to the finite element grids and assuring that
CitcomSVE calculations use the exactly same ice loads as that for analytical solutions.

**Table 3: Relative Errors for Surface 3-Component Displacement Rates for GIA Benchmark**

|  | GIA_R1 | GIA_R2 | GIA_R3 | GIA_R3B[a] | GIA_R3_LT[b] | GIA_R3_LT_SH20[c] | GIA_R3_LT_SH64 |
|---|---|---|---|---|---|---|---|
| Resolution | 48x48x48 | 48x80x80 | 64x128x128 | 64x128x128 | 64x128x128 | 64x128x128 | 64x128x128 |
| Total steps | 976 | 976 | 976 | 976 | 592 | 592 | 592 |
| # Cores | 96 | 96 | 384 | 384 | 192 | 192 | 384 |
| CPU hours | 5.57[d] | 4.89 | 3.01 | 3.13 | 3.88 | 3.34 | 3.77 |
| $\epsilon_r(0)$[e] | 17.1% (15.8%)[f] | 8.7% (8.1%) | 4.7% (4.4%) | 4.4% (3.8%) | 4.6% (4.4%) | 5.0% (4.8%) | 4.7% (4.4%) |
| $\epsilon_h(0)$ | 14.8% (15.0%) | 6.9% (6.9%) | 3.9% (3.9%) | 3.5% (3.4%) | 3.9% (3.9%) | 3.9% (3.9%) | 3.9% (3.9%) |
| $\epsilon_r(15)$ | 7.9% (6.7%) | 4.5% (4.1%) | 3.1% (3.0%) | 2.8% (2.3%) | 3.1% (3.0%) | 3.1% (3.0%) | 3.2% (3.0%) |
| $\epsilon_h(15)$ | 4.4% (3.9%) | 2.6% (2.4%) | 1.7% (1.7%) | 1.6% (1.5%) | 1.7% (1.7%) | 1.7% (1.7%) | 1.7% (1.7%) |
| $\epsilon_r(26)$ | 7.9% (6.6%) | 3.8% (3.3%) | 2.5% (2.3%) | 2.3% (1.8%) | 3.1% (3.0%) | 3.0% (2.9%) | 3.2% (3.1%) |
| $\epsilon_h(26)$ | 4.4% (3.9%) | 2.3% (2.0%) | 1.3% (1.3%) | 1.4% (1.1%) | 1.9% (1.8%) | 1.9% (1.8%) | 1.9% (1.9%) |


[a] The differences between cases GIA_R3B and GIA_R3 are discussed in section 3.2.2.
[b] The "LT" in GIA_R3_LT represents larger time increments between time steps, where the increments are
250 years and 125 years before and after 26 kybp, respectively. Cases GIA_R1, GIA_R2, and GIA_R3
have uniform time increment of 125 years.
[c] The "SH20" in GIA_R3_LT_SH20 represents that the cut-off of degrees and orders of spherical harmonics
in this calculation is 20. Similarly, case GIA_R3_LT_SH64 has cut off at degrees and orders of 64. Other
cases are cut off at degrees and orders of 32.
[d] For this case, the solution converges slowly, causing larger CPU time. All the cases are computed on the
NCAR supercomputer Derecho.
[e] $\epsilon_r$ and $\epsilon_h$ are errors of displacement rates in radial and horizontal directions, respectively. The errors are
given at present-day (0), 15 kybp, and 26 kybp (indicated by numbers inside parentheses).
[f] Numbers out of parentheses are errors calculated based on regular grids, whereas numbers inside of
parentheses are calculated based on CitcomSVE grids.



### 3.2.3. Benchmark results.


We compare the 3-component displacement rates at the surface for three different times (i.e., the
present-day, 15 kybp, and 26 kybp) obtained from CitcomSVE and the semi-analytical code. Figure 3 shows
the present-day displacement rate in vertical, eastern, and northern directions for case GIA_R3 from
CitcomSVE. Large uplift rates at the present day occur in North America, Fennoscandia, and West
Antarctica (Fig. 3a), suggesting ongoing rebound induced by ice melting since the last glacial maximum in
these regions. Horizontal displacement rates usually have much smaller amplitudes than that in radial
direction in those regions.

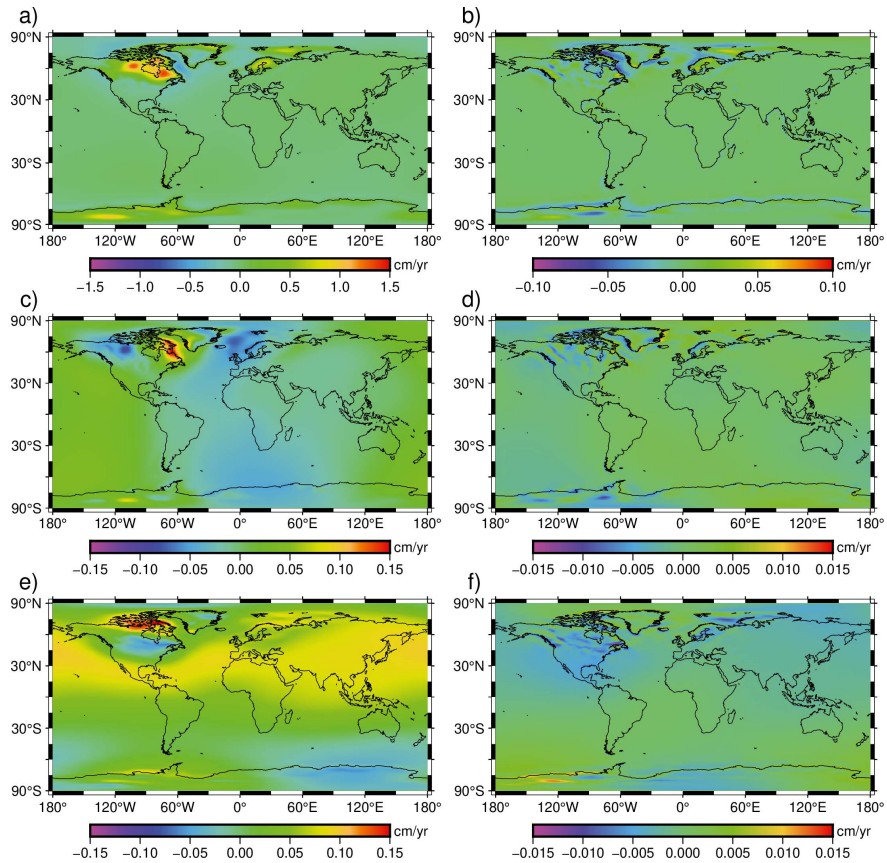





Figure 3. Displacement rates at the present day from case GIA_R3 in radial (a), eastern (c), and northern
(e) directions and the differences to analytical solutions for radial (b), eastern (d), and northern (f)
directions.

Figure 3 also shows the differences in present-day displacement rates between CitcomSVE and

semi-analytical solutions. The differences are small compared with the magnitudes of displacement rates.
Relatively large magnitudes of errors are mainly on short wavelengths (e.g. localized regions), which may
partially reflect the fact that CitcomSVE tends to have poorer accuracy at shorter wavelengths (Fig. 1 and
2). Following Zhong et al. (2022), we define relative RMS differences (i.e., errors) in displacement rates
between CitcomSVE and semi-analytical solutions as:
$$\varepsilon(t) = \sqrt{\frac{\sum[f_{FE}(\theta,\varphi,t) - f_S(\theta,\varphi,t)]^2}{\sum[f_S(\theta,\varphi,t)]^2}} \, , \qquad (16)$$
where $f_{FE}(\theta,\varphi,t)$ and $f_S(\theta,\varphi,t)$ are the fields of interest at a given time $t$ from CitcomSVE and semi-
analytical solutions, respectively, and the summation is based on a regular 1º-by-1º grid. To interpolate the
CitcomSVE solutions onto the regular grid, we use the near neighbor method provided by GMT (Wessel et
al., 2019). We also report errors calculated by unweighted summation on the CitcomSVE grid, given the
relatively uniform grid size on the spherical surface in CitcomSVE, and the differences in errors from these
two ways of calculation are insignificant. We compute errors for radial and horizontal components at three
times: the present-day, 15 kybp and 26 kybp. Note that for horizontal error, we square the difference for
each horizontal component (i.e., north and east) and add them together for each location.

Table 3 lists the errors for displacement rates at these three times for all cases, together with the

total CPU time and number of CPUs used for each case. The errors decrease significantly from GIA_R1 to
GIA_R3. For Cases GIA_R3, the errors of displacement rates are less than 5%. Case GIA_R3B, which
avoids the interpolation of the input ice loads from the regular input grid into CitcomSVE finite element
grid to eliminate the potential inconsistency in ice loads between CitcomSVE and semi-analytical
calculations, has slightly smaller errors than GIA_R3, indicating a relatively small error induced by the

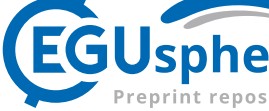

interpolation. Case GIA_R3_LT with larger time resolution before 26 kybp has larger errors in
displacement rates at 26 kybp but similar error levels at 15 kybp and present day. Those errors are close to
those from CitcomSVE 2.1 (Zhong et al., 2022). CitcomSVE 3.0 is about three times slower than
CitcomSVE 2.1 for the same resolutions since internal density variations make the computation more
expensive, as discussed in section 2.2. We found that for cases GIA_R1, GIA_R2, and GIA_R3, calculating
gravitational potential anomalies takes about one-fourth to half of the total calculation times, depending on
the time spent solving the displacement field. It is possible to speed up the calculations of the gravitational
potential anomalies by using a grid-based method (e.g., Latychev et al., 2005) or direct integration (e.g.,
Wang and Li, 2021)  for the Poisson equation instead of the currently used spherical harmonic transform.
However, the maximum degree of spherical harmonics, varying from 20 to 64, has insignificant effects on
surface displacement (Tables 3 and 4), although it would affect the modeled change rates of geoid and
gravity.

We also compare the cumulative radial displacements at different spherical harmonic degrees from

CitcomSVE and semi-analytical solutions, following previous works (Paulson et al., 2005; A et al., 2013;
Kang et al., 2022; Zhong et al., 2022). The spherical harmonic coefficients of the surface displacement field
are provided as an output of CitcomSVE (see Zhong et al., 2022, for the spherical harmonic expansion used
in CitcomSVE). The degree amplitude for each $l$ is calculated by

$$a_l(t) = \sqrt{\frac{1}{l+1}\sum_{m=0}^{l}[C_{lm}(t)^2 + S_{lm}(t)^2]} \ , \tag{17}$$

where $C_{lm}$ and $S_{lm}$ denote the cosine and sine parts of the spherical harmonic coefficients expanded from
the radial displacement fields at time $t$. Figures 4a-4c show the amplitude $a_l$ of surface radial displacement
at selected spherical harmonics degrees ($l$=1, 2, 5, 9, 16 and 23) for the three CitcomSVE cases, together
with the corresponding semi-analytical solutions. Same as CitcomSVE 2.1 (Zhong et al., 2022), the lowest-
resolution case is adequate for relatively long wavelengths ($l$=1, 2, 5, and 9), whereas higher resolution
models are required for accuracy in shorter wavelengths ($l$=16 and 23) (Fig. 4c). Figure 4d shows the results



for the harmonic of $l$=2 and $m$=1 that corresponds to the polar wander. Similar to findings from single
harmonic benchmarks in the previous section and Zhong et al., (2022), high spatial resolution is required
to obtain an accurate solution for the polar wander term. Note that the amplitudes of polar wander mode
are much smaller than other long wavelength modes like $l$=2, 5, and 9.

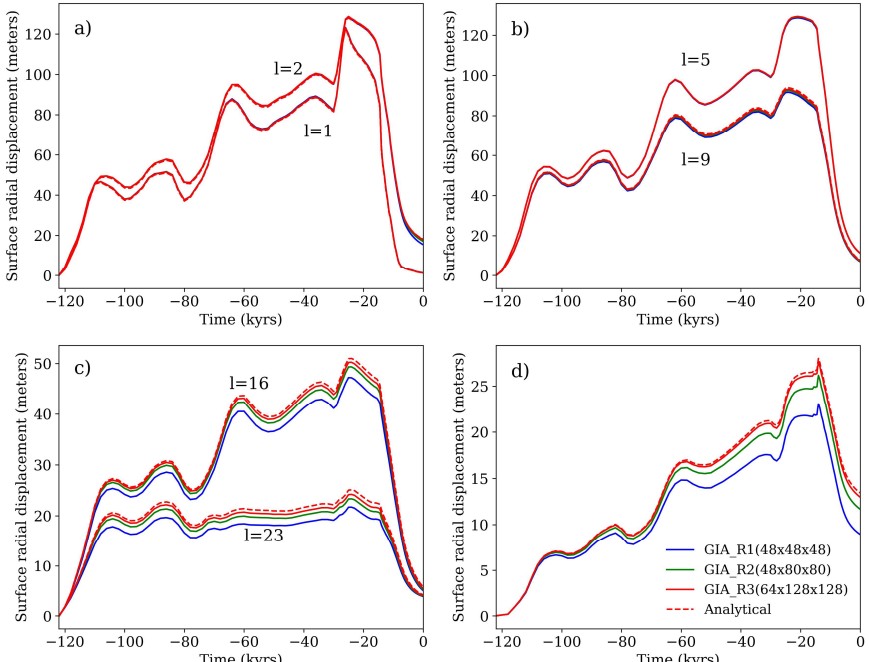


Figure 4. Amplitudes of cumulative radial surface displacement at different spherical harmonic degrees as
a function of time for the semi-analytical solutions (Analytical) and three CitcomSVE calculations
(GIA_R1, GIA_R2, and GIA_R3) for $l$=1,2 (a), $l$=5,9 (b), $l$=16, 23 (c), and polar wander mode with $l$=2,
m=1 (d).

Following Zhong et al., (2022), we use the time-integrated relative error of degree amplitude $\varepsilon_l$ to

quantify the time-averaged error for a given degree $l$. $\varepsilon_l$ is defined as

$$\varepsilon_l = \sqrt{\frac{\int_0^T [a_{l_{FE}}(t) - a_{l_S}(t)]^2 dt}{\int_0^T a_{l_S}(t)^2 dt}} \, , \tag{18}$$




where  $a_{l_{FE}}(t)$  and  $a_{l_S}(t)$  represent the degree amplitudes at time $t$ from the CitcomSVE and semi-
analytical solutions, respectively, and $T$ is the entire calculation period. The errors for each case are shown
in Table 4. As expected, the errors decrease with increasing spatial resolution for each degree, and errors
for shorter wavelengths are larger than those for longer wavelengths, except for the polar wander term with
relatively large errors.
**Table 4 Relative Errors for Surface Radial Displacements at Different Harmonics**

|  | GIA_R1 | GIA_R2 | GIA_R3 | GIA_R3_LT | GIA_R3_LT_SH20 | GIA_R3_LT_SH64 |
|---|---|---|---|---|---|---|
| $\epsilon_1$ | 0.97% | 0.74% | 0.62% | 0.64% | 0.64% | 0.64% |
| $\epsilon_2$ | 0.98% | 0.76% | 0.73% | 0.74% | 0.74% | 0.72% |
| $\epsilon_5$ | 0.33% | 0.12% | 0.13% | 0.14% | 0.14% | 0.14% |
| $\epsilon_9$ | 2.30% | 1.37% | 0.77% | 0.77% | 0.77% | 0.77% |
| $\epsilon_{16}$ | 7.56% | 3.30% | 1.45% | 1.45% | 1.45% | 1.45% |
| $\epsilon_{23}$ | 13.66% | 6.69% | 3.10% | 3.10% | N/A[b] | 3.10% |
| $\epsilon_{2,1}$[a] | 17.53% | 6.58% | 1.48% | 1.39% | 1.39% | 1.80% |

[a] $\epsilon_{2,1}$ represents the errors for the polar wander term (l=2, m=1).
[b] N/A, the cut-off of degrees and orders of spherical harmonics is 20 for this case and we only output the
spherical harmonics up to the cut-off value in CitcomSVE.

Figure 5 shows the comparisons of modeled relative sea levels at different periods (5 kybp, 10

kybp, and 15 kybp) for GIA_R3 and the semi-analytical solutions on map views. The globally averaged
relative misfits at 5 kybp, 10 kybp, and 15 kybp are 4.14%, 2.82%, and 1.70%, respectively, similar to
errors in displacement rates. The regions with localized, relatively large errors (Fig. 5b, 5d, and 5f) are
mostly around the edges of ice sheets in North America, Fennoscandia, and Antarctica, similar to that for
displacement rates, as shown in Figure 3b. Figure 6 compares modeled RSL curves for several sites from
semi-analytical solutions and three CitcomSVE calculations with different spatial resolutions. Increasing
spatial resolution reduces the misfits to semi-analytical solutions for near-field sites (i.e., sites close to ice
sheets) (Fig. 6a and 6b), but does not appear to affect the far-field solutions as much (Fig. 6c and 6d).

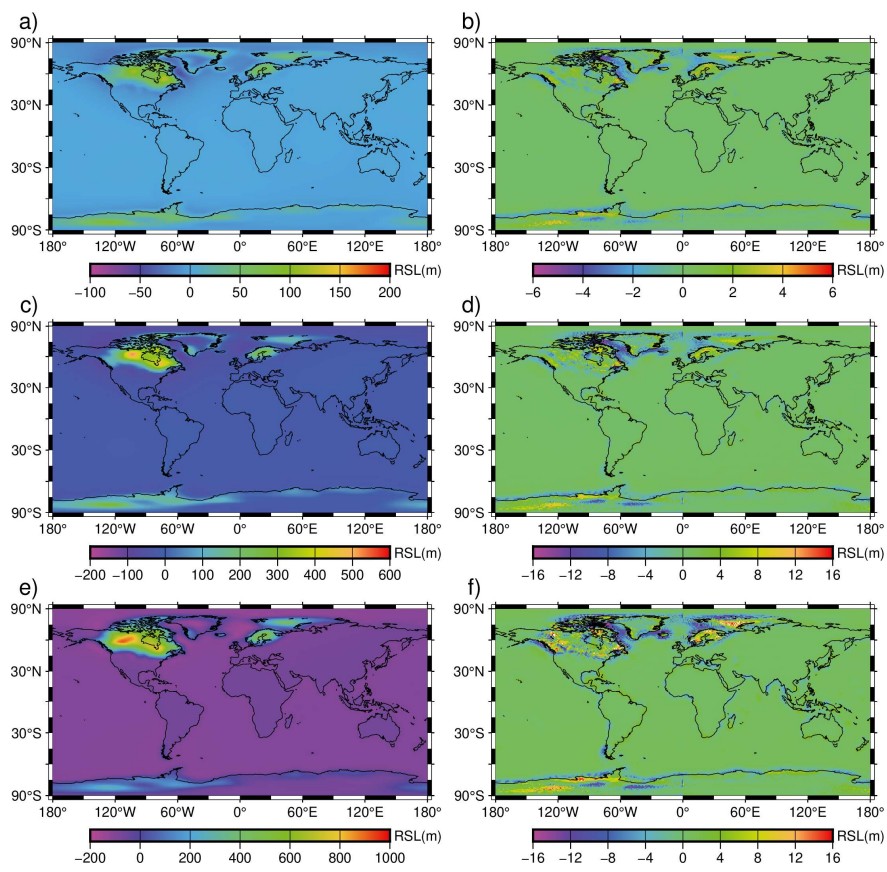


Figure 5. Map of modeled relative sea level at 5 kybp (a), 10 kybp (c), and 15 kybp (e) from GIA_R3 and
their differences to semi-analytic solutions at 5 kybp (b), 10 kybp (d), and 15 kybp (f), respectively.

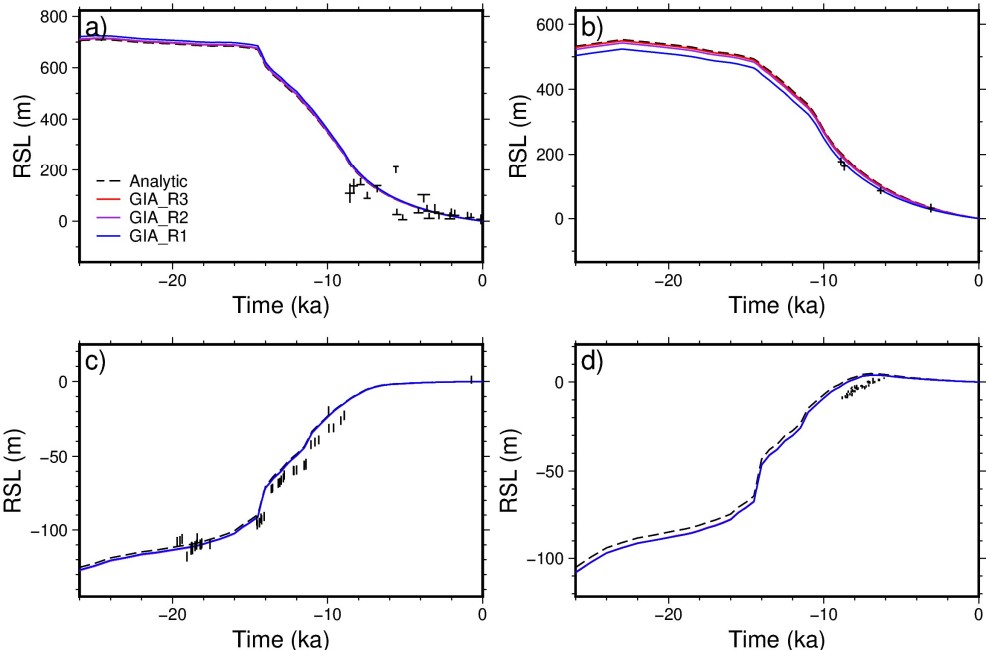

Figure 6. Relative sea-level curves for the last 26 ky at four sites from semi-analytic solutions (Analytic) and three CitcomSVE calculations of different resolutions: cases GIA_R1, GIA_R2, and GIA_R3. The four sites are Churchill (a), Vasterbotten (b), Barbados (c), and Geylang (d) with longitudes and latitudes of (265.60, 58.70), (19.90, 64.00), (300.45, 13.04), and (103.87, 1.31), respectively. The symbols represent the observed RSL changes. The observed RLS are from Peltier et al., (2015) and Lambeck et al., (2014).

## 4. Conclusion and Discussion

This study introduces CitcomSVE-3.0, an enhanced finite element package that builds upon its predecessor, CitcomSVE-2.1 (Zhong et al., 2022), an efficient package that utilizes massively parallelized computers with up to thousands of CPUs. The new version incorporates elastic compressibility (e.g., the PREM) based on the work of A et al. (2013) and improves the algorithm for solving sea level equations following the work of Kendall et al. (2005), which considers the changes in ocean loads and ocean functions related to ocean-continent transitions and the existence of floating ice. Two benchmark problems are



computed with different numerical resolutions: 1) surface loads of different single harmonics and 2) GIA problem with ICE6G ice model.

Extensive comparisons between CitcomSVE-3.0 calculations and semi-analytic solutions are presented to validate the accuracy of the CitcomSVE package. The accuracy of CitcomSVE with a horizontal resolution of ~ 50 km is better than 0.1% up to spherical harmonics of degree 4 and better than 2% up to degree 16 in vertical motion and gravitational potential for single harmonic loading problems. We show that CitcomSVE has a second order of accuracy, i.e., the errors would be reduced to 1/4 if element sizes were reduced by a factor of two. For GIA problems with realistic ice models and dynamically determined ocean loads, the average errors for CitcomSVE models with ~50 km horizontal resolution are less than 5% in displacement rates and relative sea levels.

As shown in the benchmark work for CitcomSVE-2.1 (Zhong et al., 2022), CitcomSVE has a parallel computation efficiency of > 75% for up to 6144 CPU cores. With its accuracy and efficiency in modeling viscoelastic response to surface loads and tidal forces, the open-source package CitcomSVE has the ability to advance research in planetary and climatic sciences, including GIA-related problems.

**Acknowledgement:** This work is supported by NSF through grant numbers NSF-EAR 2222115 and NSF-OPP 2333940. Our calculations were performed on parallel supercomputer Derecho operated by the National Center for Atmospheric Research under CISL project codes UCUD0007.

**Code and Data Availability Statement:** The updated CitcomSVE package can be downloaded from https://github.com/shjzhong/CitcomSVE. The input files and results to produce figures and tables for this



study can be downloaded from https://doi.org/10.5281/zenodo.13932411 (Yuan, 2024). The original ICE-
6G ice history model is from https://www.atmosp.physics.utoronto.ca/~peltier/data.php.

**Author contribution**:   All authors contributed to the development of the code, design of the research,
analysis of the results, and writing of the manuscript. T.Y. performed numerical calculations.

**Competing interests**: The authors declare there are no competing interests in this work.



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
