# Peer review of "1. Introduction"

_EGUsphere, 2024_

## Referee Comment (RC1)

**Review of egusphere-2024-3200**

by Tao Yuan, Shijie Zhong, and Geruo A

The authors present the publication and benchmark of the open source FE software package CitcomSVE 3.0, which allows to solve the GIA problem for a viscolestic continuum with lateral variations in material poperties considering elastic compressibility and the usual requirements for a GIA solver which are rotational deformations due to polar wander, geocenter motion and the sea level equation.

They benchmarked the code against a spectral 1D code following a similar benchmark of the incompressible precursor. 2.1.

The method to solve the equations for a compressible continuum with CitcomSVE2.1 was already presented by A et al. (2013) but without the SLE solver of the incompressible version and so lacking a comparable benchmark for GIA problems. Due to lack of suitable 3D benchmarks the authors were forced to test their model against the established spectral normal mode theory for 1D problems. This is in agreement with the testing of further 3D codes. To my knowledge only Martinec 2000 presented a benchmark against an analytical not spherical symmetric solution.

In summary, the presented method sounds reasonable and the results show a rather good agreement with the provided 1D solutions. Nevertheless I have a small number of suggestions which might improve the discussion and the reliability of the code:

1.  discussion of spectral loads at least up to d/o 128,

2.  transfer of the indepth discusssion of the applied new SLE solver into the supplement,

3.  discussion also of the geoid displacement for the GIA example.

Otherwise this paper is set up clearly and I suggest its consideration for GMD.

Volker Klemann

**Details**

L3 Although discussed in the paper the applicability to solve the GIA problem is not stated in the title.

L34 not clear if also compressibility can vary laterally.

L38 Is the SLE solver is part of published software?

L40 Only at the end I found an explanation of what a second-order accuracy means. But, I am not convinced if this criterion holds heres see there.

L42 An assessment of the computation time is given. May be you can add that it is three times slower than the incompressible version

L70 You should add here Tanaka et al. (2011, doi:10.1111/j.1365-246X.2010.04854.x), where like in A et al. (2013), compressibility is considered. Here also see the discussion of L200ff. you should also discuss there, which codes are compressible and which are incompressible.

L115 Your code works in the Lagrangian domain. Then I would state, that the density increment is considered as being in the Eulerian domain, first as its advection in Eq. 2 is of second order, and second that in this way the Poisson equation (Eq. 3) holds. But you could also state that in case of small perturations and the resulting linearisation the Eulerian and Lagrangian density increment do not differ.

L126 The boundary condition at the CMB (Eq. 5) is important (and also goes back to Wu and Peltier 1982).

L 127 According to Zhong et al. 2003 the equation holds for an incompressilbe core.

L 132 Here and in the following I would prefer 'continuum' instead of 'medium', due the continuum mechanical formulation of the problem.

L149 Small suggestion: 'Maxwell rheology (6)' should be sufficient to write.

L156 For the time integration of the field equations you apply an explicit time differencing scheme. Is this correct? I would then specify this.

L176ff Can you state that this coincides with Tanaka et al. (2011).

L200ff You should place this discussion to L70ff.

L255ff This is a recap of Kendall et al. 2005. May be you can reduce this section and refer to them. Also in Spada and Melini (2019, doi:10.5194/gmd-12-5055-2019) a nice overview is given.
One further aspect you do not discuss is, how you treat the inner iteration between subsequent integration steps. Is this omitted here similar to Hagedoorn et al. (2007, doi:10.1007/s00024-007-0186-7), where also the field equations are solved explicitly in the time domain?

L293 The main reason to run the outer iterations is to approximate a consistent initial topography. I did not find this explicitly stated.

L298ff This first iteration is an interesting suggestions.

L304 The efficiency is not shown in the next section but later in 3.2.1. Nevertheless as stated there, I would shift this discussion to the supplement as it interrupts the benchmark discussion in this section.

L313 Why not call t his subsection 'Spectral surface load with step-function in time' ?

L317 You can also here specify that you vary the load between (1, 0) and (16, 8).

L321 Why do you considere only the cosine term and not the complete representation of the spectral load distribution?

Table 1 The reader would help if you list here the reference Maxwell time used for normalisation, also in view of Fig. 1 and the following discussion. Furthermore I wonder why the viscosity in the upper mantle is higher than in the lower mantle, this does not look like vm5a.

L335ff Can you state something regarding the radial discretisation? How many elements are considered in the lithosphere, upper and lower mantle, respectively.

With respect to the considered spectral representations did you check if the derived load love numbers deviate for different orders, I think you have checked this but it would be interesting how much they vary also in view of the spectral solutions. The reader might also wonder why the lln of (2, 1) differ so much from (2, 0). Obviously it is due to the polar motion term. You should state this here.

Figure 1. Here you chose a different nomenclature to specify the degree/order forcing. In the next you describe the (l, m) nomenclature. Easiest would be to keept it in Figure 1 but change it in Table 2 and throughout the text.

Furthermore there is a big step from degree 16 presented here to degree 128 usually considered in GIA (see Spada et al., 2011 or Tanaka et al., 2011). So it would be interesting to show the deviations also at such high degrees (see main points).

Figure 2. Form the figure and the caption it is not visible where R5 is applied. In the lower or in the upper triangle, although it should be the lower one of course.

L422 '[...] and (12) with the floating ice criterion' ?

L424 'multiple' sounds like at least 10. Also Kendall et al or other authors usually only consider 3 to 4 iterations.

L428 as stated at L304 I would shift the whole discussion of 3.2.1 to the supplement. This as you also only refer to figures there. May be you can summarise the main output there. Here it disrupts the benchmark section (see also main points).

L433 here and throughout the text I would replace 'kybp' by 'ka BP' as used in literature, see also Figure 4 vs. Figure 6.

L531 If you shifted 3.2.1 you do not have to repeat the setup of the problem here, as this as given already before.

L555 Further down you apply a nearest neighbor algorithm for the interpolation of the displacement field. Did you apply the same algorithm here or did you use a mass conserving algorithm?

Table 3. It would be great to see here also the error statistics of the RSL for the presented epochs further down.

L582 May be you can state at the beginning that in this subsection you present surface displacement rates, and RSL. What I miss is the gravity change signal at pt, as a further prominent observable (see also main points).

L627 Here you mention the gravity change and change rate of geoid height, but you do not show results, also what about RSL for this specific case?

L640ff Is the required higher resolution for (2, 1) only an observation, or can you give an explanation for this deviating behaviour? The relative difference between -120 and 0 ky is much larger for this term in comparison to the other coefficients. May be Cambiotti et al. (2010, 0.1111/j.1365-246X.2010.04791.x) helps.

Figure 6, You should discuss the offset between the FE and S solutions at the far field sites. Is this due to coastal levering especially at Geylang or mismatch in L2m1?

L691 I would not call the presented comparison 'extensive', as you discuss rather low degree spectral loads and only one GIA realisation.

L694ff I do not follow this calculus. Considering the errors in Table 3. There, from R1 to R2 the error reduces by a factor of 2, whereas you increased the number of elements by a factor of 2.7.

L699ff You can also state here that the integration time for a compressible continuum is three times larger than for the incompressible solution.

---

## Author Response (AR1)

**Reply to Review of egusphere-2024-3200 from Volker Klemann**

We thank Dr. Volker Klemann for his careful review. In the following, we respond to the comments in a point-to-point manner and the original comments from the reviewer are italicized and in blue fonts.

*The authors present the publication and benchmark of the open source FE software package CitcomSVE 3.0, which allows to solve the GIA problem for a viscolestic continuum with lateral variations in material poperties considering elastic compressibility and the usual requirements for a GIA solver which are rotational deformations due to polar wander, geocenter motion and the sea level equation.*

*They benchmarked the code against a spectral 1D code following a similar benchmark of the incompressible precursor. 2.1.*

*The method to solve the equations for a compressible continuum with CitcomSVE2.1 was already presented by A et al. (2013) but without the SLE solver of the incompressible version and so lacking a comparable benchmark for GIA problems. Due to lack of suitable 3D benchmarks the authors were forced to test their model against the established spectral normal mode theory for 1D problems. This is in agreement with the testing of further 3D codes. To my knowledge only Martinec 2000 presented a benchmark against an analytical not spherical symmetric solution.*

*In summary, the presented method sounds reasonable and the results show a rather good agreement with the provided 1D solutions. Nevertheless I have a small number of suggestions which might improve the discussion and the reliability of the code:*

*1. discussion of spectral loads at least up to d/o 128,*

*2. transfer of the indepth discusssion of the applied new SLE solver into the supplement,*

*3. discussion also of the geoid displacement for the GIA example.*

*Otherwise this paper is set up clearly and I suggest its consideration for GMD. Volker Klemann*

**Response**: We appreciate the three suggestions listed above and revised the manuscript accordingly.

First, we added a new case of spectral load of degree 64 instead of 128 for the following reasons. To calculate Love numbers for a spectral load of degree $N$, it is necessary to calculate gravitational potential up to at least degree $N$. As CitcomSVE-3.0 calculates gravitational potential in spherical harmonic domains and displacements in finite-element domains and needs transformations between those two, it becomes significantly more expensive (both in efficiency and memory requirements) with increasing maximum spherical harmonic degrees in potential field calculation. Although the maximum harmonic degree of potential calculation affects the calculated

potential and geoid, it has a much smaller effect on the surface displacement and relative sea level, as shown in Table 3. In practice, having a maximum degree larger than 32 or 64 is usually unnecessary for displacement and relative sea level calculations. So, for spectral load cases, we chose to add a case with loading at degree 64.

For spectral load of degree 64, four resolutions are tested: 12x80x128x128 (R5), 12x80x192x192 (R6), 12x80x256x256 (R7), 12x96x256x256 (R8).

The following figure shows the Love numbers from CitcomSVE-3.0 and from the semi-analytical solution (also in figure 1).

[Figure]

The results of this new case are also added into Table 2 in the revised manuscript.

Second, we simplify the discussion of the new SLE solver in the main text and put extra discussion into the supplement.

Third, we added more results and discussion for geoid calculations. For example, in the revised Table 3, we included the misfits for geoid rate at different stages (and we also added misfits for RSL). And we also added the geoid rates and its misfit in a figure along with displacement rates.

We also included benchmark results for a Heaviside tidal load case with degree 2 and order 0 in Table 2.

**Details**

*L3 Although discussed in the paper the applicability to solve the GIA problem is not stated in the title.*

We now mention GIA in the title: CitcomSVE-3.0: A Three-dimensional Finite Element Software Package for Modeling Load-induced Deformation and Glacial Isostatic Adjustment for an Earth with Viscoelastic and Compressible Mantle

*L34 not clear if also compressibility can vary laterally.*

We clarified that the compressibility (actually both $\lambda, \mu$) can vary laterally in the abstract.

*L38 Is the SLE solver is part of published software?*

Yes. It is included in CitcomSVE-3.0.

*L40 Only at the end I found an explanation of what a second-order accuracy means. But, I am not convinced if this criterion holds heres see there.*

"Order of accuracy" is commonly used in numerical analysis and computational science to quantify the rate of convergence of a numerical approximation of a differential equation to the exact solution. Fig. 2 shows the error trends to horizontal resolution for Love numbers, and the slope is about 2 on the log-log scale, indicating errors are roughly proportional to the square of the grid sizes (i.e., second-order accuracy).

*L42 An assessment of the computation time is given. May be you can add that it is three times slower than the incompressible version*

That is mentioned in the main text, and in this revision we also mentioned it in the conclusion section. The increased computation time is mainly caused by calculations of gravitational potential with the spectral method currently used in CitcomSVE.

*L70 You should add here Tanaka et al. (2011, doi:10.1111/j.1365-246X.2010.04854.x), where like in A et al. (2013), compressibility is considered. Here also see the discussion of L200ff. you should also discuss there, which codes are compressible and which are incompressible.*

We added Tanaka et al. 2011 here and discussed which codes are incompressible.

*L115 Your code works in the Lagrangian domain. Then I would state, that the density increment is considered as being in the Eulerian domain, first as its advection in Eq. 2 is of second order, and second that in this way the Poisson equation (Eq. 3) holds. But you could also state that in case of small perturations and the resulting linearisation the Eulerian and Lagrangian density increment do not differ.*

It is true that our code works in the Lagrangian domain, and the density increment in Eq.2 and Eq.3 is Eulerian density increment; there will be no problems as long as we correctly calculate density in terms of Eulerian increment when solving the equations. We have acknowledged that $\rho_1^E$ used in those equations is Eulerian density perturbation, and the definition of $\rho_1^E$ (Eq. 1) includes effects of both volume variation and advection of the initial density field, as it should for Eulerian increment. The Eulerian and Lagrangian density increments differ by an advection term, which is $u_i\rho_{,i}$, a first-order term, not a second-order term. Detailed description of $\rho_1^E$ can be found in A et al., 2013.

*L126 The boundary condition at the CMB (Eq. 5) is important (and also goes back to Wu and Peltier 1982). L 127 According to Zhong et al. 2003 the equation holds for an incompressilbe core.*

We make it clear that, in our case, the core is incompressible. Also, the core is not part of the computational domain in CitcomSVE, and the core's influence is introduced through the boundary condition. Interestingly, the analytical solutions by John Wahr that includes the core as an compressible medium and that we used here are in excellent agreement with CitcomSVE, suggesting that the core's compressibility may not play an important role.

*L 132 Here and in the following I would prefer 'continuum' instead of 'medium', due the continuum mechanical formulation of the problem.*

We adopted the suggestion.

*L149 Small suggestion: 'Maxwell rheology (6)' should be sufficient to write.*

We adopted the suggestion.

*L156 For the time integration of the field equations you apply an explicit time differencing scheme. Is this correct? I would then specify this.*

The discretization in time space is required for the Maxwell rheological equation (Eq. 6 in manuscript). As we mentioned, it is discretized in time by integrating it from $t - \Delta t$ to $t$ with a second-order trapezoid rule. The resulting formulation is essentially an implicit time differencing scheme, more specifically, a Crank-Nicolson scheme (see also Martinec 2000, Eq. 13).

*L176ff Can you state that this coincides with Tanaka et al. (2011).*

Although Tanaka et al. (2011) also used weak formulation, we are not able to find a clear similarity between the equations presented there and ours, due to significant differences in numerical methods between those two studies. However, we mentioned that Tanaka et al., 2011 also used an FE method in the introduction. Of course, our original FE formulation was in Zhong et al., (2003).

*L200ff You should place this discussion to L70ff.*

We moved this discussion into the introduction section (line 88-96).

*L255ff This is a recap of Kendall et al. 2005. May be you can reduce this section and refer to them. Also in Spada and Melini (2019, doi:10.5194/gmd-12-5055-2019) a nice overview is given.*

*One further aspect you do not discuss is, how you treat the inner iteration between subsequent integration steps. Is this omitted here similar to Hagedoorn et al. (2007,*

*doi:10.1007/s00024-007-0186-7), where also the field equations are solved explicitly in the time domain?*

We think this summary of the approach use in Kendall et al. 2005 is necessary to make our description of SLE completed, so we prefer to keep it in the main text.

The inner iteration between subsequent integration steps is similar to that of Kendall et al., 2005 and Zhong et al., 2022, where the iteration is considered converged when the changes of potential and displacement are smaller than a certain threshold. We added the description of inner iteration in lines 282-286.

*L293 The main reason to run the outer iterations is to approximate a consistent initial topography. I did not find this explicitly stated.*

This was stated in line 281: "the unknown initial topography $T_0$ needs to be determined iteratively to keep the modeled present-day topography consistent with the observed present-day topography."

*L298ff This first iteration is an interesting suggestions.*

Thanks.

*L304 The efficiency is not shown in the next section but later in 3.2.1. Nevertheless as stated there, I would shift this discussion to the supplement as it interrupts the benchmark discussion in this section.*

We follow the suggestion to put section 3.2.1 into supplementary materials.

*L313 Why not call this subsection 'Spectral surface load with step-function in time' ?*

We adopted the suggested section name.

*L317 You can also here specify that you vary the load between (1, 0) and (16, 8).*

We now describe that the tested cases range from degree 1 to degree 64 in line 322.

*L321 Why do you considere only the cosine term and not the complete representation of the spectral load distribution?*

A single real harmonic load can be either a cosine or sin term, which are essentially identical except with a phase difference. Hence, there is little difference between using either one. For simplicity, we always use the cosine term.

*Table 1 The reader would help if you list here the reference Maxwell time used for normalisation, also in view of Fig. 1 and the following discussion. Furthermore I wonder why the viscosity in the upper mantle is higher than in the lower mantle, this does not look like vm5a.*

We list reference Maxwell time in Table 1 as suggested. The viscosity in the upper mantle is $4.853 \times 10^{20}$ Pas, not $4.853 \times 10^{21}$ Pas as it was listed in the manuscript. We corrected the typo.

*L335ff Can you state something regarding the radial discretisation? How many elements are considered in the lithosphere, upper and lower mantle, respectively.*

The radial discretization was mentioned in the next paragraph. We now moved the description into this paragraph as it fits this paragraph better (around line 360).

*With respect to the considered spectral representations did you check if the derived load love numbers deviate for different orders, I think you have checked this but it would be interesting how much they vary also in view of the spectral solutions. The reader might also wonder why the lln of (2, 1) differ so much from (2, 0). Obviously it is due to the polar motion term. You should state this here.*

We make it clear that (2,1) case considers the polar wander effects at line 345. For loads of same degree but different orders, the Love numbers from numerical solutions could be slightly different, although Love numbers from semi-analytical solutions would be the same. However, it is beyond our scope to investigate this in this study.

*Figure 1. Here you chose a different nomenclature to specify the degree/order forcing. In the next you describe the (l, m) nomenclature. Easiest would be to keept it in Figure 1 but change it in Table 2 and throughout the text.*

We modified fig.1 to use l1m0 nomenclature.

*Furthermore there is a big step from degree 16 presented here to degree 128 usually considered in GIA (see Spada et al., 2011 or Tanaka et al., 2011). So it would be interesting to show the deviations also at such high degrees (see main points).*

We addressed this comment in our response to the main points above.

*Figure 2. Form the figure and the caption it is not visible where R5 is applied. In the lower or in the upper triangle, although it should be the lower one of course.*

We made it clear in the caption that "R5 has smaller relative errors compared to R4".

*L422 '[…] and (12) with the floating ice criterion' ?*

We added "floating ice criterion" here.

*L424 'multiple' sounds like at least 10. Also Kendall et al or other authors usually only consider 3 to 4 iterations.*

We made it clear here that 3-4 iterations are usually used.

*L428 as stated at L304 I would shift the whole discussion of 3.2.1 to the supplement. This as you also only refer to figures there. May be you can summarise the main output there. Here it disrupts the benchmark section (see also main points).*

We moved section 3.2.1 into supplementary materials, see line 452-456.

*L433 here and throughout the text I would replace 'kybp' by 'ka BP' as used in literature, see also Figure 4 vs. Figure 6.*

We modified the text and figures according to this suggestion.

*L531 If you shifted 3.2.1 you do not have to repeat the setup of the problem here, as this as given already before.*

We restructured this section by moving most of this paragraph into the paragraph above it (line 440-451), since it is more of a background than a direct discussion of model setup.

*L555 Further down you apply a nearest neighbor algorithm for the interpolation of the displacement field. Did you apply the same algorithm here or did you use a mass conserving algorithm?*

The interpolation here (that is, reading ice load from a regular grid and interpolating it into CitcomSVE grid) is done by (bi-)linear interpolation from regular grids to arbitrary points (i.e., irregular grids in CitcomSVE-3.0).

*Table 3. It would be great to see here also the error statistics of the RSL for the presented epochs further down.*

We added the error statistics for RSL in Table 3, see the modified Table 3 in the revised paper.

*L582 May be you can state at the beginning that in this subsection you present surface displacement rates, and RSL. What I miss is the gravity change signal at pt, as a further prominent observable (see also main points).*

We added figures for geoid rate at present day in figure 3 and the error statistics for geoid rate in Table 3, see the table and figure in the revised manuscript. And see also line 545-550 and 558-561.

*L627 Here you mention the gravity change and change rate of geoid height, but you do not show results, also what about RSL for this specific case?*

We added the results for geoid rate and error in RSL for those cases, as mentioned in our response to the main points above.

*L640ff Is the required higher resolution for (2, 1) only an observation, or can you give an explanation for this deviating behaviour? The relative difference between -120 and 0 ky is much larger for this term in comparison to the other coefficients. May be Cambiotti et al. (2010, 0.1111/j.1365-246X.2010.04791.x) helps.*

The required higher resolution for (2,1) is more of an observation and reflects the difficulty in accurately modeling polar wander, which is also discussed in A et al., 2013 and Zhong et al., 2022. The relatively large difference between -120 and 0 kr means a relatively large total displacement on degree 2 and order 1 after a full glacial cycle. Cambiotti et al., 2010 (Fig. 5-7) is informative on this topic, showing the effect of the non-hydrostatic correction ($\beta$) on total polar motion after a full glacial cycle. The detailed discussion of the nature of polar wander in ice age is beyond the scope of this study. However, it is an interesting topic that deserves future investigations.

*Figure 6, You should discuss the offset between the FE and S solutions at the far field sites. Is this due to coastal levering especially at Geylang or mismatch in L2m1?*

Figure 6 shows that the offset between FE and S solutions of RSL reduces with increased numerical resolution for near field sites, but not for far field sites. This is reasonable because the RSL at near field sites is controlled by load-induced crustal motion and potential change, whose accuracy is highly sensitive to numerical resolution, whereas RSL at far field sites is more controlled by total ice/water volume and ocean areas, which are less sensitive to numerical resolution. The offset in RSL between FE and S solutions at far field sites is not likely caused by l2m1, since the accuracy of l2m1 increases with numerical resolution. The offset is more likely related to factors other than numerical calculations, such as the interpolation of ocean function from the regular grid to the CitcomSVE grid or the interpolation of results on the CitcomSVE grid to RSL sites. We added some descriptions for the offset at far field sites in line 604-607.

*L691 I would not call the presented comparison 'extensive', as you discuss rather low degree spectral loads and only one GIA realisation.*

Now, after adding a spectral load of degree 64 and a benchmark for tidal load, and considering we have at least 4 numerical calculations with different resolutions for each

case (seven calculations for the GIA case where different scenarios were considered), we think it is fair to say the benchmark is extensive.

*L694ff I do not follow this calculus. Considering the errors in Table 3. There, from R1 to R2 the error reduces by a factor of 2, whereas you increased the number of elements by a factor of 2.7.*

The error is proportional to the grid size dx or dy to some power, not to the total number of elements. With increased number of elements by a factor of 2.7 in horizontal directions, the horizontal grid spacing is only reduced by about sqrt of 2.7 or a factor of 1.6. The second order accuracy would lead to error reduction of a factor 2.7, which is slightly larger than the actual error reduction of a factor of 2. That we did not get the full error reduction with increased resolution here could be caused by other factors in GIA calculations, for example, we did not solve the gravitational potential at the same resolution as FE grids. Also the GIA case with SLE makes determining level of accuracy more difficult since other factors affect the comparison between CitomSVE-3.0 and semi-analytical solutions, for example, the different representation of ice load and ocean functions between CitcomSVE-3.0 (on irregular grids) and the semi-analytical code (on regular grids and spherical harmonic domains).

We clarified that the accuracy discussed here is mainly from the single harmonic benchmark (spectral loads); see line 632.

*L699ff You can also state here that the integration time for a compressible continuum is three times larger than for the incompressible solution.*

We add one sentence in the last paragraph to address this comment.

**Reply to Review of egusphere-2024-3200 from Anonymous Referee #2**

We thank the anonymous referee #2 for the review. In the following, we respond to the comments in a point-to-point manner and the original comments from the reviewer are italicized and in blue fonts.

*The paper by Yuan et al. presents the next step of the GIA modelling setup using CitcomS. While the method has been published previously (SLE solver in Zhong et al. and compressibility in A et al.), here they present a benchmark of the two combinations. The paper summarizes the applied methods very well. However, I felt sometimes a bit lost in the paper as you introduce a method/set-up and then something else is mentioned afterwards. You come back to the method/set-up at a later point. I would suggest splitting the paper in two parts: 1) introducing and*

*benchmarking your model to simple harmonic loads (where no SLE is involved), and 2) the benchmark with the SLE and the ICE-6G_D ice model, which means that you introduce the SLE part after you have presented the first benchmark.*

The SLE solver presented here has some important differences compared to that published previously (Zhong et al., 2022), see section 2.3 and lines 260-264. The sea level equation is an important part of CitcomSVE-3.0, and we think it is proper to describe it in the method section. However, we follow the suggestion to put section 3.2.1 into the supplement, making the paper easier to read.

*A few other things could be done to convince the reader that your GIA modelling setup works:*

- *Show a comparison of the geoid change in the benchmark with the SLE and ICE-6G_D.*
- *Compare your radial and horizontal displacement rates as well as geoid and RSL rates to the results by Peltier et al. (which you can find on his website).*

In this revision, we add the benchmark results for geoid change in both Table 3 and Fig. 3; see those in the revised manuscript.

In this paper, we compare the results from CitcomSVE-3.0 to our semi-analytical solutions because we can ensure that all the mathematical formulations used to define the GIA problem are essentially the same for those two codes. However, those formulations used by us are not exactly the same as those used in Peltier et al., 2015, for example, the formulations for polar wander and SLE. Although the slight differences in formulations may be insignificant for modeling results, it is not ideal to compare results from different formulations for benchmark purposes. So we did not compare our results with those of Peltier et al. However, we would like to point out that our RSL results for many different sites are in excellent agreement with those computed by Glenn Milne who is one of our collaborators for another project.

*Minor comments:*

*L.73 - Huang et al. is not a coupled spectral-finite element method as it is purely based on the finite-element method. In addition, the method by Wu is also based on the finite-element method and only the potential calculations are done in the spherical harmonic domain. Not sure, if this should be then called coupled spectral-FE method. If I understand you correctly, you are doing the same by "coupling" the spherical harmonic domain with the FE method.*

We revised this and put Huang et al. 2023 and Wu 2004 into the FE method group.

*L.77 - "center of mass".*

We corrected it.

*L.144 - You mention that the Lamé parameters and viscosity can vary laterally and radially. Is this also true for the density? I couldn't find such a statement.*

The density is horizontally homogeneous. We made it clear in line 124.

*L.183 - "the applied potential, which is only relevant"*

We revised this sentence in line 192.

*L.185 - "on the unknown incremental"*

We corrected it.

*L.195-202 - Please move this paragraph somewhere else. It doesn't fit here.*

We move this paragraph into lines 89-96.

*L.212 - What "given threshold error tolerance" is implemented in CitcomSVE 2.1 & 3.0?*

The tolerance is specified in the input file for CitcomSVE. In this work, we use 0.3%. We made it clear in line 213.

*L.301 - Interesting approach. Please write it out in an equation how O(t) is calculated.*

We add a section in the supplementary text to describe further our approach for calculating $O(t)$.

*L.324 - Why do use the density and Lamé parameters from the mantle for the crust?*

It will make benchmarks easier by having simple models that can be accurately represented by both analytical and numerical codes, to avoid the differences in results caused by model differences. The crustal layer in PREM is thin, and its density and Lamé parameters change rapidly in depth, which makes it difficult to be captured by numerical models, especially for low-resolution models. We have tested it with analytical code and found that this change of properties for this thin crustal layer does not affect the results significantly. For those reasons, we ignored this thin layer and treated it as the mantle material right below it for benchmark purpose.

*Section 3.2.1 - You refer only to figures in the supplement. Either move the text to the supplement and summarize the results in the main article or move figures from the supplement to the main manuscript. I would prefer the former.*

We adopted the suggestion and put this section into the supplement.

*Section 3.2.1 - Just out of curisosity: Could you please plot K2 in Fig. S1 as well? So, the results of the 2nd iteration based on the approach by Kendall et al. No need to do this for the other figures.*

We added K2 into Fig. S1. Note that K2, K3 and AS2 almost overlap with each other.

[Figure]

*L.419 - In the abstract you say ICE-6G_D, but here it is ICE-6G now. Please use the same and correct abbreviations throughout the text.*

We changed it to ICE-6G_D throughout the paper.

*L.546 - Please use more representative names for your model types, e.g. GIA_R1 could be GIA_135 to indicate that this model has a resolution of 135 km and the reader doesn't have to go back and check what R1 was referring to.*

Although model resolutions for R1, R2, and R3 were described here within a paragraph, it is easier to find the model resolutions from Table 3, which contains information about horizontal, vertical, and time resolutions for all models. Since those models (R1, R2, R3...) also differ in vertical resolution, we are afraid that using GIA_135 as the name may mislead readers to think that the vertical resolutions are same for those models. Since we only have three different spatial resolutions (R1, R2, and R3), and they are ordered with increasing resolution, this naming convention makes it clear that R1 has the lowest spatial resolution, whereas R3 has the highest spatial resolution. Considering those reasons, we prefer this kind of naming convention.

*Table 3 - I find it difficult to compare the CPU hours when you use different amounts of cores. Please unify this to see the direct difference.*

We added a new row as core-hours (i.e., "number of CPUs" multiples runtime) in Table 3.

*Fig.S2/S3 - Use the same abbreviations for both figures.*

We modified the legend in Fig. S3 to make the abbreviations consistent with other figures.

*Figures – Copernicus recommends the authors to use colorblind-friendly colors/colormaps, which is stated on the GMD website (https://www.geoscientific-model-development.net/submission.html#figurestables). This includes avoiding the parallel usage of red and green as well as the avoiding jet and rainbow colormaps. The link above provides examples where you can find suitable colormaps and find colors for your line plots that are suitable.*

Thanks for pointing out the colormap issues. We modified Fig. 1 and Fig.4 to avoid the parallel usage of red and green, and modified Fig. 3 and Fig. 5 for another colormap.